# Defective mitochondria remodelling in B cells leads to an aged immune response

Marta Iborra-Pernichi[1,2], Jonathan Ruiz García[1,2], María Velasco de la Esperanza[1,2], Belén S. Estrada[1,2], Elena R. Bovolenta [1,2], Claudia Cifuentes[3], Cristina Prieto Carro[3], Tamara González Martínez[1,2], José García-Consuegra[4], María Fernanda Rey-Stolle [5], Francisco Javier Rupérez [5], Milagros Guerra Rodriguez [6], Rafael J. Argüello [7], Sara Cogliati[4], Fernando Martín-Belmonte [1,2] & Nuria Martínez-Martín [1,2] ✉

The B cell response in the germinal centre (GC) reaction requires a unique bioenergetic supply. Although mitochondria are remodelled upon antigen-mediated B cell receptor stimulation, mitochondrial function in B cells is still poorly understood. To gain a better understanding of the role of mitochondria in B cell function, here we generate mice with B cell-specific deficiency in Tfam, a transcription factor necessary for mitochondrial biogenesis. Tfam conditional knock-out (KO) mice display a blockage of the GC reaction and a bias of B cell differentiation towards memory B cells and aged-related B cells, hallmarks of an aged immune response. Unexpectedly, blocked GC reaction in Tfam KO mice is not caused by defects in the bioenergetic supply but is associated with a defect in the remodelling of the lysosomal compartment in B cells. Our results may thus describe a mitochondrial function for lysosome regulation and the downstream antigen presentation in B cells during the GC reaction, the disruption of which is manifested as an aged immune response.

B cells are critical players in the adaptive immune response. They are responsible for generating high-affinity antibodies and a memory cell compartment that will remain in the organism to monitor for new threats[1-4]. The T cell-dependent B cell response is a multi-step process characterised by an extrafollicular response followed by the establishment of a mature germinal centre (GC) reaction. First, the extrafollicular response generates short-lived plasmablasts (SLPBs) and memory B cells (MBCs). Then, activated B cells seed the GC reaction and undergo class-switch recombination, somatic hypermutation, and affinity maturation, leading to the production of MBCs and long-lived plasma cells (LLPCs) that produce high-affinity, class-switched antibodies[5,6]. However, new findings have challenged our current understanding of how the adaptive immune response works. For example, class-switch recombination can occur before establishing a mature GC (mat-GC)[7], and somatic hypermutation as well as affinity maturation can also occur at the extrafollicular response[8,9]. Moreover, there is evidence that many (and possibly most) MBCs are formed in the extrafollicular response stage[10-13], suggesting that only the GC is required to develop LLPCs[10,14].

[1]Program of Tissue and Organ Homeostasis, Centro de Biología Molecular "Severo Ochoa", Consejo Superior de Investigaciones Científicas-Universidad Autónoma de Madrid, Madrid, Spain. [2]Intestinal Morphogenesis and Homeostasis Group, Area 3-Cancer, Instituto Ramón y Cajal de Investigación Sanitaria (IRYCIS), Madrid, Spain. [3]Program of Interactions with the Environment, Centro de Biología Molecular "Severo Ochoa", Consejo Superior de Investigaciones Científicas-Universidad Autónoma de Madrid, Madrid, Spain. [4]Program of Physiological and Pathological Processes, Centro de Biología Molecular "Severo Ochoa", Consejo Superior de Investigaciones Científicas-Universidad Autónoma de Madrid, Madrid, Spain. [5]Centre for Metabolomics and Bioanalysis (CEMBIO), Facultad de Farmacia, Universidad San Pablo-CEU, CEU Universities, Madrid, Spain. [6]Electron Microscopy Facility, Centro de Biología Molecular "Severo Ochoa, " Consejo Superior de Investigaciones Científicas-Universidad Autónoma de Madrid, Madrid, Spain. [7]Aix Marseille Univ, CNRS, INSERM, CIML, Centre d'Immunologie de Marseille-Luminy, Marseille, France. ✉e-mail: nmartinez@cbm.csic.es

Establishing an extrafollicular response and a subsequent GC reaction requires the collaboration of multiple cell types[15], in which the interaction between B and T cells is critical[16,17]. Naïve cognate CD4 T cells are primed by dendritic cells, triggering CD4 T cell differentiation into a T follicular helper cell fate ($T_{fh}$)[16,17]. B cells recognise a cognate antigen in soluble form or on professional antigen-presenting cells[18]. Antigen recognition by the B cell receptor (BCR) leads to its internalisation and degradation into lysosome-like structures from which antigen-peptides are loaded into MHCII complexes[19]. Finally, these MHCII-peptide complexes are exposed for T cell receptor recognition on the plasma membrane. After this first activation, CD4 T cells (pre-$T_{fh}$ cells) and pre-activated B cells migrate towards the T:B border where B cells present MHCII-peptides to pre-$T_{fh}$. This interaction with pre-$T_{fh}$ provides B cells with survival and co-stimulatory signals critical for their full activation based on the interaction of CD40-CD40L, among others. These two activated cells form motile conjugates that maintain stable contact for minutes to hours[20–22]. T-B conjugates migrate towards the centre of the follicle, where activated B cells differentiate, a process that starts by establishing a pre-GC and ends with the mat-GC stage.

At the molecular level, antigen recognition by B and CD4 T cells triggers a precise set of transcriptional and epigenetic changes. This leads to the expression of a common set of regulators (GC modulators) whose expression is reinforced by establishing T-B conjugates[23–31]. One of these modulators is BCL6[32], whose expression is indispensable for establishing and maintaining an effective GC reaction. BCL6 protein expression in B and $T_{fh}$ cells is interconnected, and insufficient BCL6 in T cells leads to impaired CD40L signalling to B cells[33], which, in turn, leads to the extinction of BCL6 expression and hence GC dissolution.

In vivo, naïve B cells are quiescent, while GC B cells (as activated B cells) are highly proliferative and metabolically active[34–36]. Activated B cells at the GC undergo metabolic rewiring in response to stimulation by follicular dendritic and $T_{fh}$ cells but also in response to hypoxia[37]. This metabolic rewiring is driven by the activation of the PI3K-Akt-mTORC1 axis[36,38] and seems coupled with an increase in both mitochondria-dependent (tricarboxylic acid (TCA) and oxidative phosphorylation (OxPhos)) and -independent (aerobic glycolysis) metabolism[36,38]. Interestingly, GC B cells have recently been described as relying primarily on fatty acid oxidation (FAO)[34], which is puzzling since GC B cells have higher metabolic demands in a hypoxic environment and are expected to be glycolytic instead. Consequently, new studies to further elucidate GC's metabolic profile are needed.

In lymphocytes, antigen-triggered metabolic rewiring has traditionally been linked to mitochondria remodelling. In B cells, antigen encounter is intimately associated with ATP generation to sustain the bioenergetic demands of B cells[36,39]. Mitochondria are linked to myriad functions in immune cells, e.g. they can act as a signalling platform through the production of reactive oxygen species (ROS) or intermediate metabolites[40], shape calcium signalling, trigger apoptosis, control protein translation and export peptides or mitochondrial DNA (mtDNA)[41]. These functions may require the interplay between mitochondria and other organelles, such as the endoplasmic reticulum, peroxisomes, and lysosomes[42]. Hence, mitochondria can be considered a central hub for the activation and proliferation of lymphocytes as well as for fate decisions and certain effector functions.

Mitochondrial dysfunction has been described to underlie ageing in immune cells[43,44]. The multifaceted nature of mitochondria requires a closely controlled status, content, and interconnection with other organelles, which are achieved through molecular pathways that shape mitochondrial biogenesis, fusion/fission, and mitophagy after antigen encounter. In this context, mitochondria remodelling in B cells has been described to drive the production of specific intermediate metabolites (immunometabolites) such as haem, phosphatidic acid, and α-ketoglutarate, which can shape the expression of BCL6 and BLIMP-1 intrinsically, thus instructing B cell fate differentiation[38,45].

Most studies examining the role of mitochondria in B cells utilised genetically modified mice for molecules targeting the PI3K-Akt-mTORC1 axis[36,38]. While this pathway shapes mitochondrial biogenesis in B cells and many other processes[46], the main question remains unanswered: What functions do mitochondria have in activated B cells during the GC reaction?

To answer this, we generate a B cell-specific knock-out (KO) that targets mitochondrial function by deleting mitochondrial transcription factor A (Tfam), a nuclear-encoded transcriptional factor controlling mtDNA copy number and expression. Tfam knock-out (KO) mice demonstrate a disruption in the germinal centre (GC) reaction, resulting in an aged immune response. Notably, the compromised GC reaction in *Tfam* KO mice is ascribed to a deficiency in the remodelling of lysosomal compartments in B cells, consequently impacting antigen presentation. As a result, our research unveils a function for mitochondria in B cells, laying the groundwork for establishing a connection between mitochondrial dysfunction in B cells and the aging of the immune response.

## Results

### Tfam drives B cell mitochondria remodelling upon antigen encounter

To investigate the specific role of mitochondria in B cell function, conditional *Tfam*^fl/fl-targeted mice[47] were crossed with *mb1*^Cre mice expressing Cre recombinase under the promoter of the *mb1* gene[48]. Using fluorescence-activated cell sorting (FACS), the B cell compartment in *Tfam*^fl/fl *mb1*^Cre- (wild-type, WT) and *Tfam*^fl/fl *mb1*^Cre+ (KO) mice was characterised. The absence of *Tfam* significantly reduced the percentage and absolute number of total B2 B cells (Fig. 1A and Suppl. Fig. 2D) in the spleen. However, there was no dramatic change in the frequency of marginal (B220+ CD93- IgM^high CD21^high) and follicular (B220+ CD93- IgM+ CD21^low) cells, showing only a slight increase in the former (Fig. 1A and Suppl. Fig. 2B–D). Also, there was a slight increase in the frequency of total transitional B cells (B220+ CD93+) (Suppl. Fig. 2A). There were no differences in the percentage and development of B1 B cells in the peritoneal cavity (Suppl. Fig. 3).

The reduction in the frequency of total mature B2 B cells in the spleen was concurrent with a blockage in B cell development in the bone marrow (Suppl. Fig. 4). *Tfam* KO mice presented half the frequency and absolute number of immature cells and a threefold lower amount of pro-pool cells (Suppl. Fig. 4A), suggesting a blockage in early B cell development. A deeper analysis of pro-pool B cells showed a blockage in the transition from pro-to pre-B cells (Suppl. Fig. 4B). Furthermore, a dramatic reduction in recirculating B cells was observed (Suppl. Fig. 4A), suggesting that Tfam plays a crucial role in bone marrow development in the pro-B cell stage, where metabolic checkpoints have been described previously[49]. However, mature B cells still populate the spleen and the peritoneal cavity in *Tfam* KO mice, showing normal differentiation and expressing similar levels of IgM and IgD compared to their WT counterparts (Suppl. Figs. 2, 3).

To exclude the possibility of splenic B cells escaping Cre-recombinase action[45], total DNA, RNA, and protein were extracted from naïve WT and KO-sorted B cells. The expression levels of *Tfam* (Fig. 1B and Suppl. Fig. 5A), mtDNA-encoded genes (*MT-ND1, MT-CO1, MT-ATP6*), and the amount of mtDNA (Suppl. Fig. 5B) were quantified by qPCR. Results show that *Tfam* KO B cells lack specifically *Tfam* (Fig. 1B). As expected, while KO B cells display normal levels of genomic encoded genes, they presented diminished levels of mtDNA copies and reduced expression of mtDNA-encoded genes (Suppl. Fig. 5B). Finally, CD4 and CD8 T cells (Fig. 1B) showed normal levels of *Tfam*. These results demonstrate the specificity of our conditional mouse model.

It is known that B cell stimulation is concomitant to an increase in mitochondria content, i.e. mitochondria biogenesis, accompanied by a burst in OxPhos and glycolysis rate to meet the

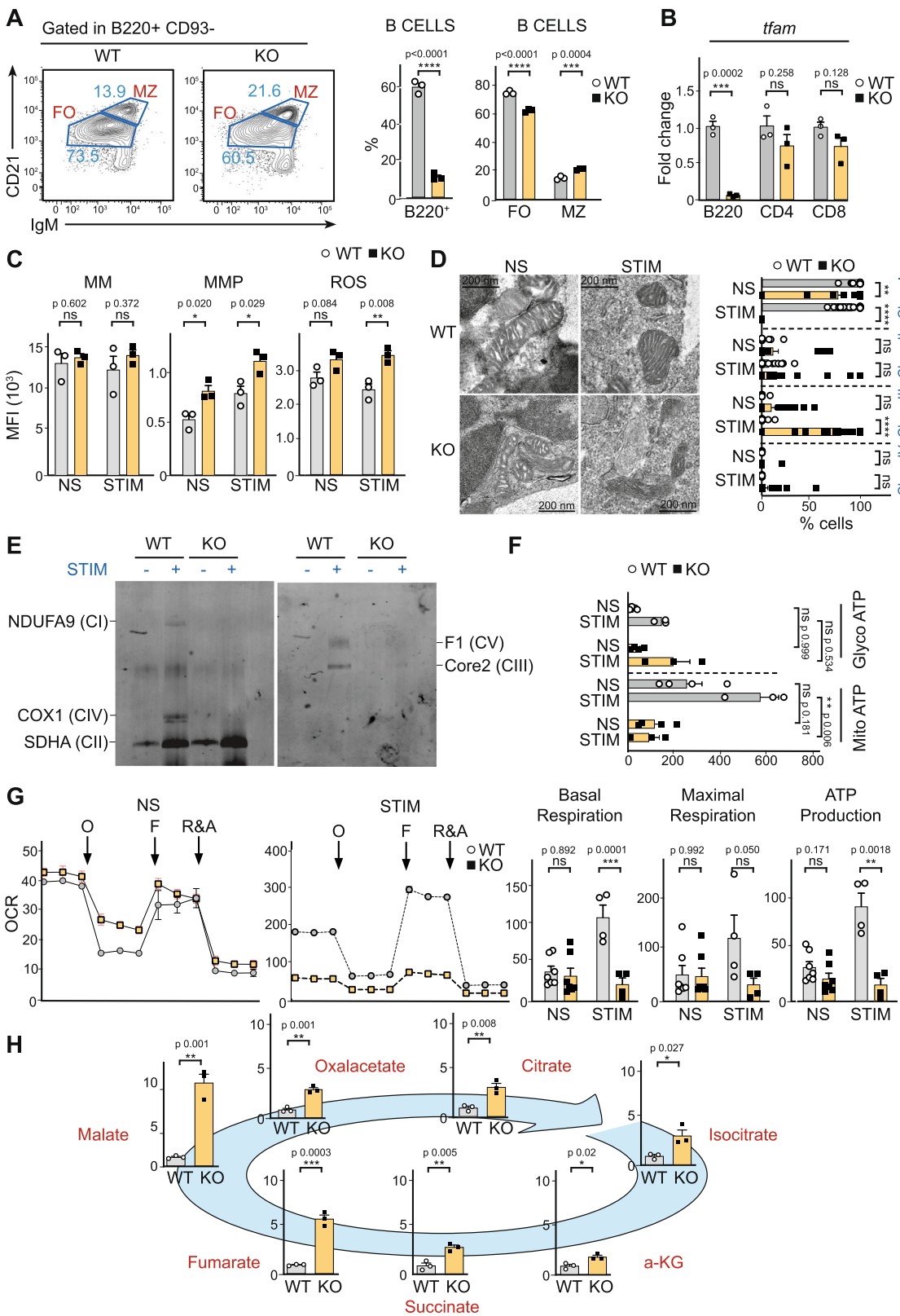

bioenergetic requirements of activated B cells[38]. To examine whether Tfam could be a master regulator of these processes in B cells, we investigated the mitochondrial mass and membrane potential of non-stimulated (naïve) and 24-h-stimulated B cells (anti-IgM and anti-CD40). *Tfam* KO B cells only showed a slight increase in mitochondrial membrane potential compared with WT B cells (Fig. 1C), suggesting that the lack of *Tfam* does not drastically impact

mitochondrial content after antigen encounter in B cells. Using the CM-H2DCFDA probe, a general oxidative stress indicator to monitor ROS, an increase in total ROS production in KO B cells after stimulation was observed (Fig. 1C).

Next, mitochondrial ultrastructure was analysed by electron microscopy in purified, non-stimulated, and 24- h-stimulated WT and KO B cells (Fig. 1D and Suppl. Fig. 6A). As indicated by the

**Fig. 1 | *Tfam* deletion in B cells leads to failure in mitochondrial remodelling.**
**A** Left, flow cytometry plots showing marginal (MZ, IgM$^{high}$,CD21$^{high}$) and follicular (FO IgM$^+$, CD21$^{low}$) zone within the B2 cell population (B220$^+$CD93$^-$) in WT and *Tfam* KO mice. Right, the percentage of total B cells (gated as B220$^+$) and B2 cells (FO, MZ). **B** Relative *Tfam* mRNA levels determined by RT-PCR in sorted B220$^+$, CD4$^+$, and CD8$^+$ T cells from WT and *Tfam* KO mice. **C** Mitochondrial status. From left to right: bar charts indicating mean fluorescence intensity (MFI) values of Mitotracker Green (MM), Mitogreen red CMXROS (MMP), and total cellular ROS (ROS) quantified by flow cytometry. WT and *Tfam* KO B cells were purified and unstimulated (NS) or 24 h stimulated with anti-CD40 and anti-IgM (STIM). **D** Left, representative electron microscopy fields of WT and *Tfam* KO B cells in pre- (NS) and post-stimulation (STIM) conditions. Right, a bar chart showing the percentage of mitochondria assigned to morphological classes I–III as described in Scorrano et al. Morphometric analysis was blinded and performed on randomly selected electron microscopy fields from three independent experiments. **E** Immunodetection of the indicated proteins after Blue Native page of dodecyl maltoside (DDM) -solubilized mitochondria from WT and *Tfam* KO cells pre- and post-stimulation with anti-CD40 and anti-IgM. Each sample is a pool of at least six mice per group (WT and KO non-

stimulated and 24 h stimulated) and is a representative experiment from two independent experiments. **F** Quantification of ATP rate derived from Mitochondrial respiration and Glycolysis in non-stimulated (NS) or 24 h stimulated (STIM) purified B cells from WT and *Tfam* KO mice. **G** Left, a representative oxygen consumption rate (OCR) graph was obtained from extracellular flux assay in NS and STIM purified B cells. The OCR was measured before and after injection of oligomycin (O), FCCP (F), and rotenone plus antimycin A (R&A). Right, bar charts indicating levels of basal respiration, maximal respiration, and ATP production of NS and STIM purified B cells from WT and *Tfam* KO mice. **H** Bar charts indicating the relative amount of compounds found in the TCA cycle in cell lysates of 24 h-stimulated B cells from WT and *Tfam* KO mice. In panels 1**A**–**C**, **F**–**H**, bar charts show the quantification of one representative experiment out of three with at least three biological replicates, and error bars represent mean ± SEM (Additionally, in panel **H**, each dot represents one mouse and is the average of three technical replicates). Panel **B** and **D** two-way ANOVA, with additional Tukey's multiple comparisons test, was conducted. For the rest of the panels, an unpaired two-tailed *t*-test was conducted. *P* values: *$p < 0.05$, **$p < 0.01$, ***$p < 0.001$ and ****$p < 0,0001$. See also Suppl. Figs. 2–8. Source data are provided as a Source Data file.

morphometric analysis after stimulation, KO B cells accumulated mitochondria with dramatic morphological derangement including disorganised cristae and partial rupture of the outer membrane (class III and IV)[50] (Fig. 1D). Possibly to counteract mitochondrial derangement, KO B cells showed increased mitochondrial mass (Suppl. Fig. 6B).

Since Tfam controls replication and transcription of mtDNA-encoded genes, we examined whether Tfam could affect the amount of electron transport chain complexes after BCR stimulation. Blue native polyacrylamide gel electrophoresis showed that WT B cells, after stimulation, had upregulated CI, CII, CIII, CIV and CV (Fig. 1E). However, KO cells were unable to increase the level of mtDNA-encoded complexes upon stimulation (Fig. 1E). By contrast, KO B cells displayed increased levels of nuclear-encoded (CII) complex upon stimulation. These data suggest that Tfam is a master regulator of mitochondria remodelling after B cell stimulation.

To further elucidate the consequences of abrogated mitochondria remodelling in B cells, mitochondrial and glycolytic ATP production rates were characterised using a Seahorse XFe96 Flux analyser (Agilent Technologies) to monitor the oxygen consumption rate (OCR) and extracellular acidification rate in non-stimulated and stimulated WT and *Tfam* KO B cells. Both cell types showed similar glycolytic ATP production rates and there was only a slight (non-statistically significant) reduction in mitochondrial ATP production in non-stimulated cells (Fig. 1F). Upon activation with anti-IgM and anti-CD40, WT B cells upregulated the glycolytic ATP production rate and, more importantly, the mitochondrial ATP production rate (Fig. 1F). In contrast, KO B cells upregulated only the glycolytic rate (Fig. 1F). This failure in upregulating mitochondria ATP production rate is clearly evidenced with the OCR profiles obtained for both cell pools (Fig. 1G and Suppl. Fig. 7C). In non-stimulated conditions, WT and KO cells were almost equivalent (Fig. 1G and Suppl. Fig. 7A, C), but *Tfam*-defective cells failed to upregulate mitochondrial activity after stimulation (Fig. 1G and Suppl. Fig. 7B, C), remaining in a naïve-like mitochondrial status (Suppl. Fig. 7C).

An untargeted metabolomic analysis (liquid chromatography-mass spectrometry) was performed to compare glycolytic and TCA intermediates in stimulated WT and *Tfam* KO B cells. A slight but significant increase in intracellular glucose, phosphoglycerate, and lactate content was found (Suppl. Fig. 8). All detected TCA intermediates showed a general increase (Fig. 1H). Since TCA and OxPhos rates are linked, this widespread accumulation can be due to a decelerated OxPhos activity concurrent with a naïve-like mitochondrial phenotype in KO B cells (Fig. 1F, G and Suppl. Fig. 7). Moreover, the accumulation of TCA intermediates is more dramatic downstream of succinate dehydrogenase (i.e. fumarate and malate) (Fig. 1H), consistent with the

upregulation of this enzyme in KO cells (Fig. 1E). Together, we conclude that Tfam is a master regulator of mitochondria remodelling upon antigen encounter in B cells.

## Mitochondria remodelling drives the transition from pre-GC to mat-GC

To evaluate the impact of *Tfam* deficiency in B cells upon antigen challenge, WT and KO mice were immunised intraperitoneally with sheep red blood cells (SRBCs). Spleens were analysed on day 7 post-immunisation (p.i.) by flow cytometry. Expectedly, we found that the percentage of total B cells in KO mice remained reduced compared with WT mice (Suppl. Fig. 9A). While WT mice showed ~4% of GC B cells (B220$^+$GL7$^+$CD95$^+$), KO mice exhibited a severe blockage resulting in only half the frequency of GC cells (Fig. 2A). These results were confirmed through inspection of splenic sections using confocal microscopy (Suppl. Fig. 9B) which showed reduced follicle area (B220$^+$) and a dramatic reduction in the number of GC structures (GL7$^+$) per follicle in KO mice (Suppl. Fig. 9C). Moreover, due to significant inhibition of class switching recombination, KO mice presented a halving in the percentage of switched GC cells (GL7$^+$CD95$^+$IgD$^-$) (Suppl. Fig. 9D).

To exclude the possibility of a defective GC reaction due to a reduced number of B cells in the periphery, we aimed to delete specifically *Tfam* in GC B cells. For this purpose, we crossed conditional *Tfam$^{fl/fl}$*-targeted mice[47] with *Cγ1$^{Cre}$* mice expressing Cre recombinase under the promoter of the *IGHG1* locus[51]. *Tfam$^{fl/fl}$ Cg1$^{Cre-}$* (WT) and *Tfam$^{fl/fl}$ Cγ1$^{Cre+}$* (KO$^{Cγ1}$) were immunised with SRBCs. The *Tfam* deletion was analysed on day 7 p.i. (Suppl. Fig. 10). For this purpose, total DNA and RNA were extracted from follicular B cells (FO), GC WT and KO $^{Cγ1}$ sorted B cells. qPCR quantification showed that expression levels of *Tfam*, mtDNA-encoded genes (*MT-CYB*) (Suppl. Fig. 10A), and the amount of mtDNA (Suppl. Fig. 10B) dropped specifically in GC cells in KO$^{Cγ1}$ mice demonstrating the specificity of this conditional mouse model. Through flow cytometric analysis of spleens, KO$^{Cγ1}$ mice exhibited, as *Tfam$^{fl/fl}$ mb1$^{Cre+}$* (Fig. 2), a defect in the GC reaction, showing a reduction in the frequency of total GC cells (B220$^+$GL7$^+$CD95$^+$) (Suppl. Fig. 10C), and supporting the fact that *Tfam* deletion is responsible for the failure in GC reaction upon antigen encounter.

For the *Tfam$^{fl/fl}$ mb1$^{Cre}$* model, at the GC, KO mice displayed a drastic and specific reduction in the percentage of centrocytes (GL7$^+$CD95$^+$CXCR4$^-$CD86$^+$) (Fig. 2B). The absence of a normal centrocyte population in KO mice was accompanied by the appearance of an intermediate population with low levels of CD86 (Fig. 2B). Furthermore, centrocytes did not show increased levels of caspase 3 in *Tfam* KO mice, thus excluding the possibility of an increased apoptosis rate in this population (Suppl. Fig. 9E). However, centroblasts in KO

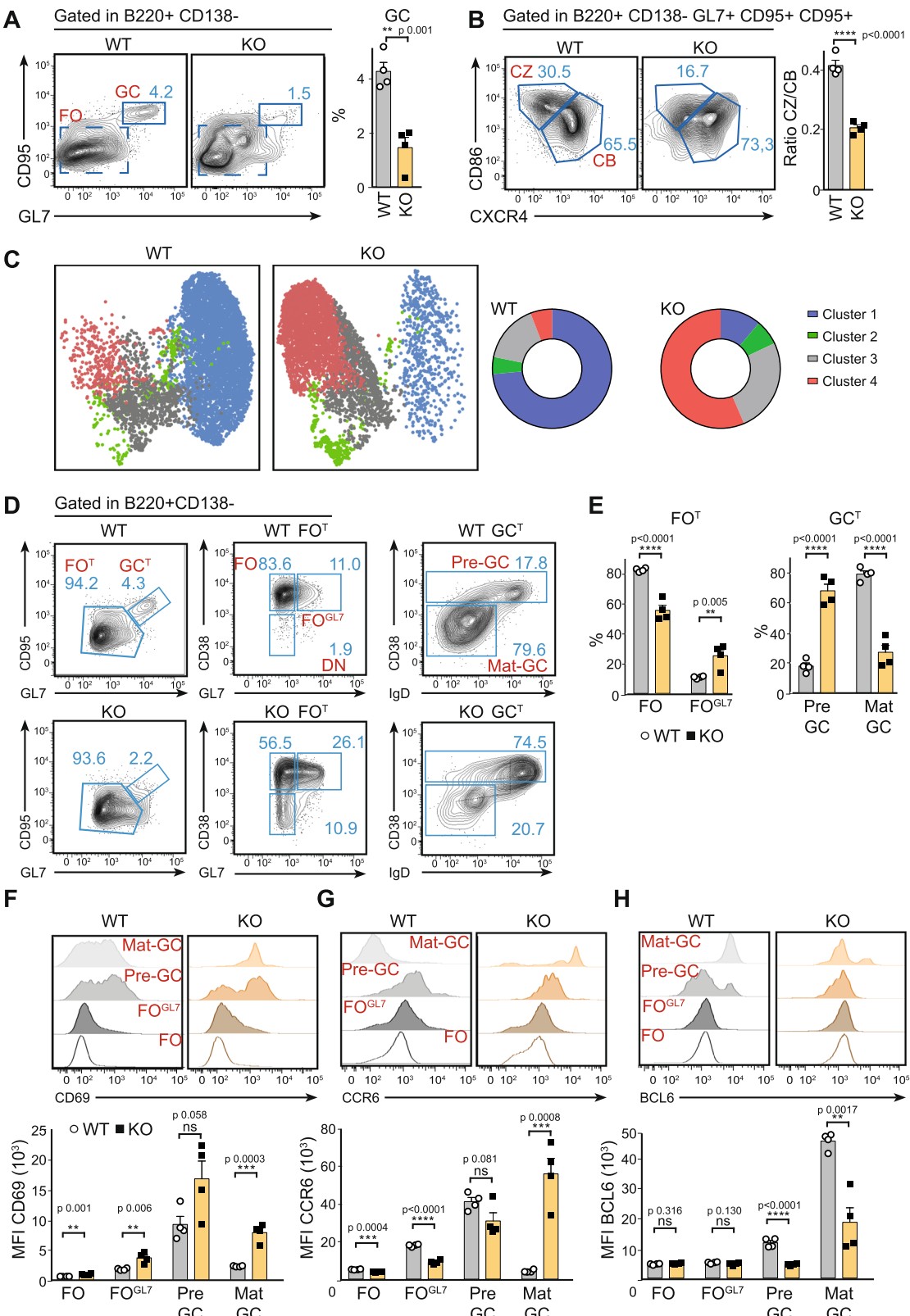

mice showed decreased levels of Ki67, suggesting that fewer cells might egress from this population (Suppl. Fig. 9E). Overall, *Tfam* KO mice presented a defective GC reaction which demonstrates the relevance of mitochondria remodelling for its establishment.

To address the stage of GC maturation at which mitochondria remodelling is indispensable, a more thorough analysis of the GC reaction was conducted by designing an antibody panel that included

proteins involved in GC maturation, such as BCL6, CCR6, IgD, CXCR4, and CD38[52–54]. Spleen cells from immunised mice were analysed using FACS at day 7 p.i. (SRBCs). Data from B220+/CD95+/GL7+ cells (identified as total GC cells) were subjected to dimension reduction using the UMAP algorithm. Clustering of the whole GC population using the FSOM clustering tool yielded 13 clusters according to Elbow analysis (Suppl. Fig. 11A–C). To simplify the study (i.e. the number of clusters),

**Fig. 2 | Failure in mitochondrial remodelling leads to a blockage at the pre-GC stage. A** Left, flow cytometry plots showing splenic germinal centre (GC, CD95[+], GL7[+]) and FO (CD95[−], GL7[−]) B cell populations 7 days post-immunisation (p.i.). Right, corresponding bar charts indicate the proportion of GC B cells (gated in B220[+]CD138[−]). **B** Left, flow cytometry analysis of the principal populations of GC indicated as centrocytes (CZ, CD86[hi]/CXCR4[lo]) and centroblasts (CB, CD86[lo]/CXCR4[hi]) of WT and *Tfam* KO mice. Right, a graph showing the CZ/CB ratio in GC 7 days p.i. **C** Unsupervised uniform manifold approximation and projection (UMAP) and FlowSOM algorithms were used in OMIQ, which yielded distinct clusters in WT and *Tfam* KO mice 7 days p.i. The left panel shows the UMAP distribution of up to 3.000 downsampled events based on CD38, IgD, Bcl6, CCR6 and CXCR4 markers and gated manually on total GC (B220[+] CD138[−] GL7[+] CD95[+]). The right panel shows charts of the proportion of clusters generated by FSOM analysis in WT and *Tfam* KO samples. **D, E** Flow cytometry plots showing (left to right) total FO (FO[T]) and total GC (GC[T]) gated in B220[+]CD138[−] cells. FO (CD38[+], GL7[−]), FO[GL7] (CD38[+], GL7[+]) and DN (CD38[−], GL7[−]) gated in FO[T]. Pre-GC (CD38[+]) and Mat-GC (CD38[−]IgD[−]) gated on GC[T]. **E** Bar charts indicate the proportion of each cell population gated on panel (**D**). Data were analysed using two-way ANOVA and are representative of three independent experiments. **F–H** Histograms and quantification of activation markers CD69 (**F**), CCR6 (**G**) and Bcl6 (**H**) in cell populations gated on panel (**D**) were analysed by flow cytometry. See Suppl. Fig. 1A for gating strategy. In all panels, bar charts show the quantification of one representative experiment out of three, with at least three biological replicates and error bars represent mean ± SEM. Each dot represents one mouse. For panel 2**E** two-way ANOVA, with additional Tukey's multiple comparisons test. For the rest of the panels, unpaired two-tailed *t*-test: *$p < 0.05$, **$p < 0.01$, ***$p < 0.001$ and ****$p < 0,0001$ See also Suppl. Figs. 9–11. Source data are provided as a Source Data file.

FSOM analysis was limited to four clusters based on CD38 levels, a known tracer for GC maturation (Suppl. Fig. 11D). Results show that GC B cells from control mice accumulated in cluster 1, a region characterised by BCL6[hi], IgD[−], CCR6[−], and CD38[−], all features of a mat-GC (Fig. 2C and Suppl. Fig. 11E). In contrast, GC B cells from *Tfam* KO mice accumulated in cluster 4 which is defined by markers such as CD38[+], IgD[+] and BCL6[lo], features of immature GC (Fig. 2C and Suppl. Fig. 11E). Notably, there are transitional stages (clusters 2 and 3) with intermediate levels of IgD and BCL6. GC from KO mice accumulated in cluster 3, which is defined by lower levels of IgD than cluster 4 but cannot upregulate BCL6 (Fig. 2C and Suppl. Fig. 11E).

A gating strategy was designed to verify the OMIQ results observed in *Tfam* KO mice (Fig. 2D). All FO cells (FO[T]) were divided into two populations based on their GL7 and CD38 levels: FO (B220[+], CD138[−], GL7[−], CD95[−]) and FO[GL7] (B220[+], CD138[−], GL7[int], CD95[−]) (Fig. 2D). Similarly, GC cells (GC[T], B220[+]CD138[−]GL7[high]CD95[+]) were divided into pre-GC (CD38[+]) and a mat-GC (IgD[−]CD38[−]) populations based on CD38 levels (Fig. 2D). Pre- and mat-GC populations were found to coincide with UMAP clusters 4 and 1, respectively (Suppl. Fig. 11F, G).

FO, FO[GL7], pre-GC and mat-GC were tested as possible gating strategies to monitor B cell activation and GC maturation (Fig. 2F–H). CD69 and CCR6 levels in each population in the WT mice showed that FO[GL7] coincided with early activated cells and exhibited elevated levels of CD69 (Fig. 2F) and CCR6 (Fig. 2G) compared with FO. CD69 and CCR6 increased and peaked during the pre-GC stage before decreasing during the mat-GC stage (Fig. 2F, G). BCL6, the master regulator of the GC differentiation programme, was upregulated at pre-GC and reached a maximum at mat-GC (Fig. 2H). BCL6 has been described as a repressor of CCR6 expression[55], which would explain the mutually exclusive expression between BCL6 and CCR6 at the mat-GC stage (Fig. 2G, H).

By applying this strategy, B cells with defective *Tfam* were found to lead to a blockage during the early stages of activation, more precisely during the FO[GL7] and pre-GC stages, resulting in only a small percentage of B cells reaching the mat-GC stage (Fig. 2D, E). *Tfam* KO and WT cells at FO[GL7] and pre-GC stages showed comparable CD69 and CCR6 levels (Fig. 2F, G), suggesting they are activated normally. However, in *Tfam* KO mice, BCL6 upregulation was completely abrogated in the transition from FO[GL7] to pre-GC (Fig. 2H), and there was a drastic inhibition in the transition to mat-GC which impeded full maturation of B cells, explaining the blockage at the pre-GC stage (Fig. 2D, E). Mat-GC showed higher levels of CCR6 in *Tfam* KO than controls, which might be due to the lack of BCL6 upregulation[55] (Fig. 2G, H).

Co-stimulatory molecule expression (CD40, ICOSL) and MHCII levels were assessed throughout GC maturation (Suppl. Fig. 12A). With the exception of a slight impairment in ICOSL expression at the pre-GC stage, *Tfam* KO cells did not exhibit any significant deficiencies in these parameters that could account for the observed complete blockage in GC maturation.

Cumulatively, results show that *Tfam* deletion leads to a blockage at the pre-GC stage due to a defect in BCL6 upregulation, which abrogates GC maturation.

## Tfam deletion in B cells triggers metabolic plasticity to maintain bioenergetic homoeostasis

Mitochondria remodelling has been proposed in B cells as an indispensable mechanism to provide ATP and bioenergetically sustain B cell activation, proliferation, and differentiation at the GC reaction[36]. To test whether *Tfam* deletion leads to a defect in ATP homoeostasis in activated B cells, the relative intracellular ATP content was quantified by FACS in non-stimulated and stimulated B cells in vitro and during GC maturation in vivo by monitoring quinacrine signal[56] (Fig. 3A and Suppl. Fig. 12B)[56]. *Tfam* KO B cells display slightly decreased levels after stimulation in vitro (Suppl. Fig. 12B, C). However, compared to the control, *Tfam* KO B cells showed comparable relative ATP levels during GC maturation, displaying only a slight increase at FO and FO[GL7] stages (Fig. 3A).

To monitor overall bioenergetic status, mTORC1 activity was evaluated by measuring the phosphorylation of ribosomal protein RPS6, a substrate of the p70RPS6 kinase (an mTORC1 effector). Both non-stimulated and stimulated KO cells maintained a normal bioenergetic status comparable to control B cells in vitro (Suppl. Fig. 12D). Moreover, B cells did not show any relevant difference in mTORC1 signalling during GC maturation (Fig. 3A) apart from a slight increase in mTORC1 in FO and mat-GC stage, which correlates with higher levels of pAkt (Suppl. Fig. 12E). These results indicate that defective GC in *Tfam* KO mice (Fig. 2 and Suppl. Figs. 9–11) cannot be attributed to defects in bioenergetics and nutritional status.

To investigate bioenergetic-independent mechanisms that could explain defective GC in *Tfam* KO mice, RNAseq analysis was performed on in vitro stimulated B cells from WT and KO mice (Fig. 3B and Supplementary Data 1). The transcriptomes of WT and KO B cells were analysed by gene set enrichment analysis (GSEA) of defined mitochondrial pathways (MitoCarta 3.0[57]). Results showed suppression of OxPhos and mitochondrial complexes (Fig. 3B) but no other dysregulated mechanisms related to mitochondrial pathways that could explain the observed GC failure.

Recently, studies have described the impact of *Tfam* deletion in the transcriptomics of macrophages[58] and astrocytes[59]. In both cases, *Tfam* deletion led to functional defects in these cells due to a bioenergetic-independent mechanism, triggering metabolic rewiring as a compensatory mechanism and leading to lipid metabolism dysregulation. Transcriptomics of B cells and macrophages were compared to explore the possibility of finding a similar mechanism. For this purpose, a ¨macrophage signature¨ (i.e. a custom gene set containing differentially expressed genes in *Tfam* KO macrophages[58] compared with WT) was generated. Comparison using GSEA showed similarities between these models (Fig. 3C), suggesting that a likely metabolic adaptation could also occur in *Tfam* KO B cells. Another GSEA analysis

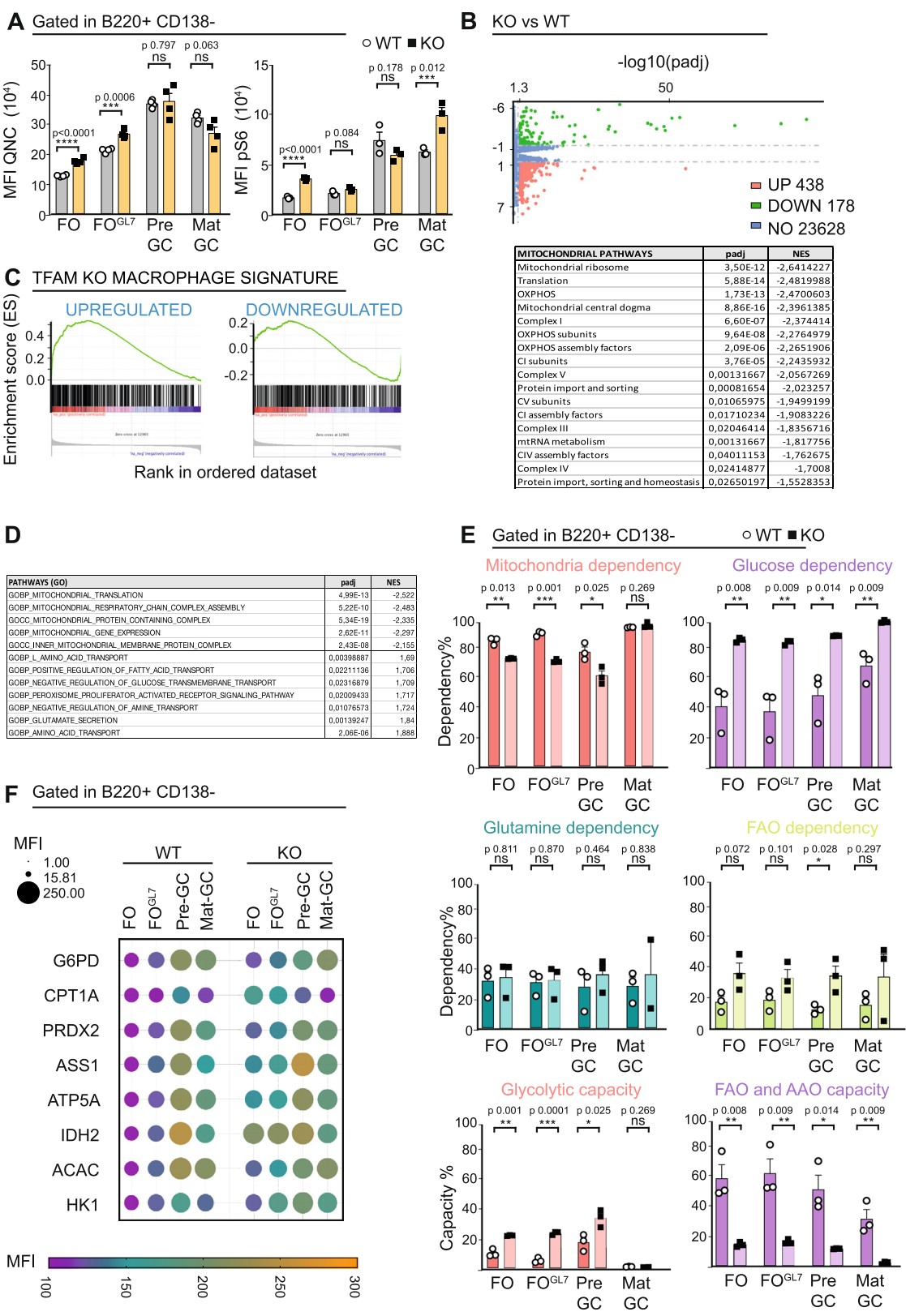

using gene ontology (GO) terms focused on metabolism confirmed a reduction in mitochondrial pathways but also changes in other metabolic pathways related to glucose, amino acids, and fatty acids (Fig. 3D). Based on these results, we hypothesised that GC maturation might correlate with metabolic remodelling from pre- to mat-GC that KO B cells could not perform properly, which could explain the blockage.

To assess metabolic remodelling in B cells during GC maturation, the SCENITH™ technique[60] was used to profile splenocyte metabolism from day 7 p.i. WT and KO mice (SRBCs). This technique is based on metabolism-dependent translation rates. By using the specific inhibitors oligomycin, 2-DG, etomoxir, and CB-839, the dependence on mitochondria, glucose, FAO, and amino acid oxidation (AAO), respectively, can be estimated from puromycin incorporation

**Fig. 3 | *Tfam* deletion in B cells triggers metabolic plasticity to maintain bioenergetic homoeostasis. A** Graphs indicating the levels of total ATP upon incubation with quinacrine (left) and levels of pS6 protein (right) during GC maturation in WT and *Tfam* KO mice 7 days p.i. **B** Volcano plot represents the distribution of differentially expressed genes (DEGs) between *Tfam* KO and WT B cells 24 h post-stimulation using the Wald test. Adjusted *p* values (*p*adj) were calculated using the Benjamin−Hochberg correction. The dotted lines bound the minimal absolute log fold-change (logFC) of 1 for the most differentially expressed genes. Red dots represent significantly upregulated DEGs with a *p*adj value <0.05 and a logFC >1, totalling 438. Green dots represent significantly downregulated DEGs with a *p*adj value <0.05 and a logFC <−1, totalling 178. Blue dots represent genes falling under the threshold. On the bottom, a table lists mitochondria-related, significantly depleted pathways defined by Gene Set Enrichment Analysis (GSEA) using the Mitocarta 3.0 database. Normalised enriched scores (NES) according to Kolmogorov−Smirnov GSEA statistics and adjusted *p* values (padj) with Benjamini−Hochberg correction are included. **C** Gene set enrichment analysis (GSEA) plots for *Tfam* KO B cells vs WT B cells compared to *Tfam* KO macrophages

vs their WT counterparts. Diagrams depict the similarity in the enrichment subset of upregulated and downregulated genes between *Tfam* KO vs WT B cells and *Tfam* KO vs WT macrophages. **D** Table obtained from GSEA for Gene Ontology terms (GO) of the differential gene expression data from *Tfam* KO vs WT B cells using the biological processes (BP) and cellular components (CC) gene sets. Normalised enrichment scores (NES) according to Kolmogorov−Smirnov GSEA statistics and *p*adj (corrected using the Benjamini−Hochberg method), are listed. **E** Metabolic profile of B cell populations along GC maturation (FO to mat-GC) obtained by SCENITH as a result of incubation with different inhibitors in WT and *Tfam* KO mice 7 days p.i. **F** Metflow profile of B cell populations along GC maturation (FO to mat-GC) obtained by the flow cytometry analysis of eight metabolic enzymes on WT and *Tfam* KO 7 days p.i. In panels 3**A** and **E**, bar charts show the quantification of one representative experiment out of three with at least three independent experiments, and error bars represent mean ± SEM. Each dot represents one mouse. Unpaired two-tailed *t*-test: *$p < 0.05$, **$p < 0.01$, ***$p < 0.001$ and ****$p < 0,0001$. See also Suppl. Figs. 12−14. Source data are provided as a Source Data file.

detected by flow cytometry (Suppl. Fig. 13A, C). Moreover, the glycolytic and FAO/AAO capacity values can be obtained. GC maturation in WT showed constant mitochondria, FAO, and AAO dependences (Fig. 3E and Suppl. Fig. 13C). A slight increase in glucose dependency correlating with increased glycolytic capacity from FO to pre-GC was found. Notably, transition from pre- to mat-GC triggered a switch-off of glycolytic capacity and reduced FAO and AAO capacities (Fig. 3E), indicating that mat-GC cells present less metabolic plasticity at this stage and supporting the idea of a metabolic rewiring from pre- to mat-GC.

*Tfam* KO GC B cells present a static metabolic profile upon antigen encounter. They are unmodified from the non-stimulated stage (FO), indicating that these cells do not remodel mitochondria function (Fig. 3E). Indeed, *Tfam* KO GC B cells present a lower mitochondria dependence that is compensated by higher dependences on glucose and FAO combined with a higher glycolytic capacity in the FO, FO[GL7], and pre-GC stages (Fig. 3E). The metabolic adaptation of B cells lacking *Tfam* was also evidenced in in vitro experiments (Suppl. Fig. 13A, B).

The higher glucose and FAO dependences coincide with increased GLUT1 expression and changes in glucose incorporation (Suppl. Fig. 14A, B), a compensatory mechanism that may allow KO cells to maintain energetic and nutritional homoeostasis according to ATP and mTORC activity levels (Fig. 3A). Nevertheless, *Tfam* KO pre-GC cells can perform the unique metabolic rewiring detected by SCENITH in WT GC maturation: switching off their glycolytic capacity (Fig. 3E).

Changes in metabolic dependencies and capacities could be due to the reprogramming of specific metabolic pathways. The Met-Flow technique[61] is a flow cytometry-based method that examines the metabolic state of cells by evaluating expression of critical proteins across the metabolic network (Suppl. Fig. 13D). Using this method (Fig. 3F), GC maturation was observed to be concomitant to induction from FO to pre-GC as indicated by HK1, G6PD, IDH2, ATP5A, ASS1, CPT1A and ACAC proteins, of the capacity for flux through glycolysis, oxidative pentose phosphate pathway, TCA cycle, OxPhos, arginine synthesis, FAO, and fatty acid synthesis, respectively. Moreover, an increase in the antioxidant response pathway, according to PRDX2 levels, was detected (Fig. 3F). The transition from pre- to mat-GC parallels a reduction of expression of the proteins mentioned above, especially those involved in the TCA cycle and FAO, mirroring a more 'quiescent' metabolic state and correlating with the reduction in their glycolytic capacity. This data highlights the different metabolic requirements of B cells during GC maturation.

KO GC maturation displayed a general upregulation of all evaluated pathways, exemplifying the metabolic adaptation of these B cells already at the naïve stage (FO) (Fig. 3F). This is also true for PRDX2, which may result from high oxidative stress produced by the

dysregulated OxPhos. But still, KO GC maturation showed a clear metabolic rewiring, especially the shift from pre- to mat-GC.

An increase in CPT1A and IDH2 in FO and FO[GL7] stages was the relevant change when comparing WT and KO cells (Fig. 3F), which correlated with increased FAO dependency identified by SCENITH (Fig. 3E). Metabolomic analysis of KO B cells stimulated in vitro showed an increase in cholesterol compared with stimulated WT cells (Suppl. Fig. 14C). We speculated whether KO B cells could have a compromised FAO, driving lipid droplet (LD) accumulation and aberrant lipid profile as occurs in macrophages and astrocytes[58,59]. To test this, lipid metabolism was characterised in WT and KO B cells in vitro and in vivo. Using a Seahorse XFe96 Flux analyser, stimulated KO B cells exhibited increased endogenous fatty acid catabolism (etomoxir-sensitive OCR) but decreased fatty acid-induced OCR (Suppl. Fig. 14D, E). LD accumulation was monitored along with GC maturation in WT and KO mice using NILE-RED (Suppl. Fig. 14F). Results showed that while KO had an accumulation of these organelles in FO and FO[GL7], KO and WT mice reached equivalent levels of LD in pre- and mat-GC stages, correlating with comparable levels of lipid peroxidation (Suppl. Fig. 14G). These results suggest that deletion of *Tfam* in B cells induces a metabolic compensation partially characterised by changes in lipid metabolism; however, LD accumulation seems not to be the leading cause of GC blockage in *Tfam* KO mice.

Overall, metabolic profiling indicates that GC maturation requires remodelling of the GC metabolic state. B cells possessed robust metabolic plasticity to maintain bioenergetic homoeostasis even with a forced "naïve-like mitochondria profile" resulting from the lack of *Tfam* (Suppl. Fig. 7C). Thus, the inability of KO B cells to mount an effective immune response must be due to mitochondria playing a specific, yet still unknown, role upon antigen encounter that differs from merely serving as ATP powerhouses.

### Mitochondria remodelling controls lysosome function

B cell interaction with cognate follicular T[fh] modulates BCL6 upregulation[33,62]. After antigen challenge in WT mice immunised with SRBCs (day 7 p.i.), a defined T[fh] population (CD4[+], CD44[+], CXCR5[+], PD1[+]) was observed. However, this population is halved in *Tfam* KO mice (Fig. 4A). Moreover, *Tfam* KO T[fh] cells expressed lower levels of BCL6 (Fig. 4B), indicating a defect during B cell activation.

To evaluate whether mitochondria remodelling after antigen encounter could shape antigen presentation in B cells, the response of WT and KO B cells in co-culture with OTII CD4[+] T cells was assessed. Carboxyfluorescein succinimidyl ester (CFSE)-labelled WT and KO B cells were stimulated with anti-IgM and ovalbumin-coated microspheres and co-cultured with CellTrace™ Violet (CTV)-labelled OTII T cells. After three days, B and T cell proliferation was assessed using flow cytometry according to CFSE and CTV dilution, respectively. KO B

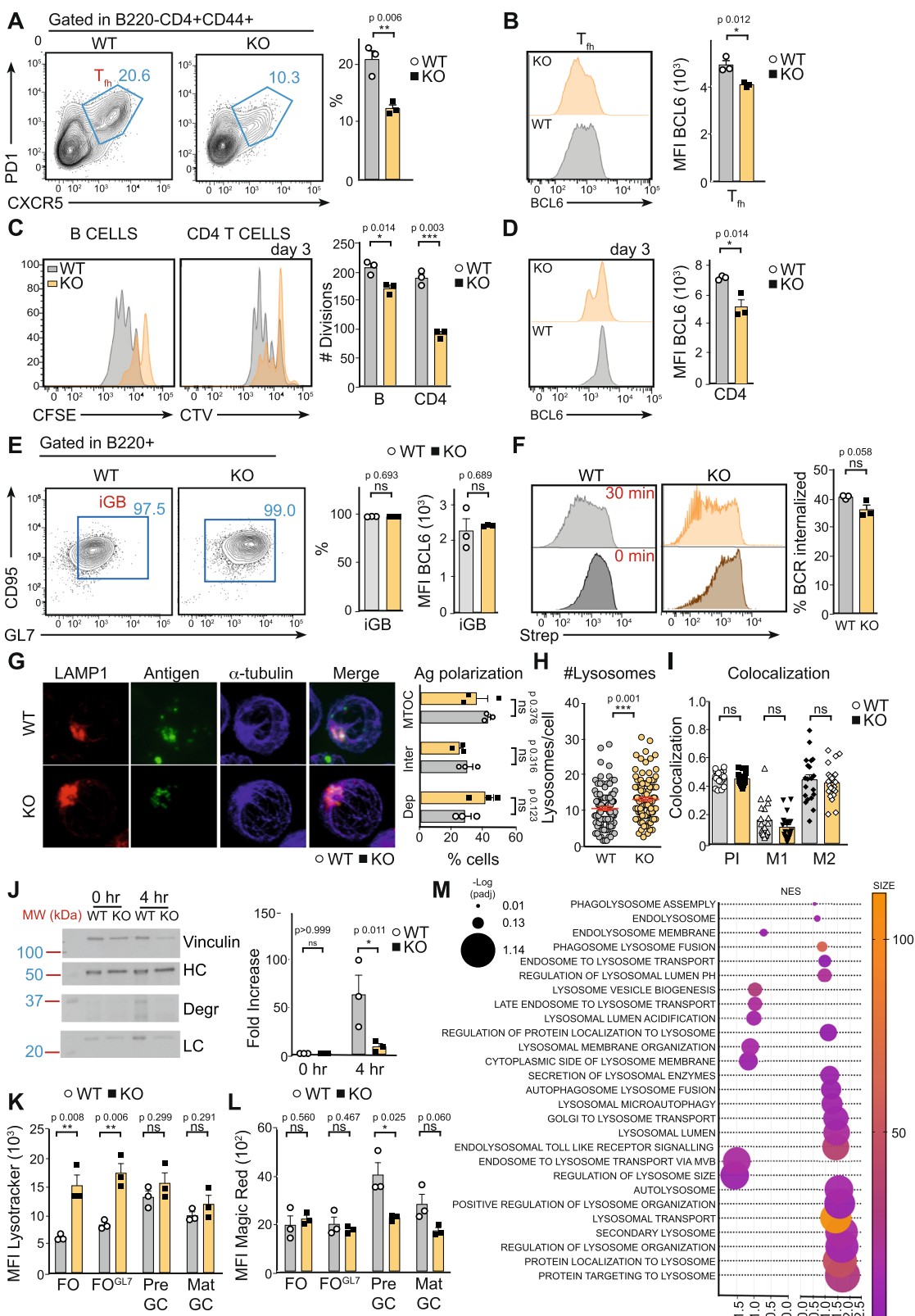

cells exhibited reduced proliferation and triggered less T cell proliferation compared with control B cells (Fig. 4C). Reduced levels of CD69 accompanied the defect in T cell proliferation (Suppl. Fig. 15A), and there was a significant inhibition of BCL6 induction on day 3 (Fig. 4D). These differences in proliferation, activation, and BCL6 expression in CD4 T cells were not due to defects in B-T conjugation formation (Suppl. Fig. 15B), and were not evident when co-cultures

were treated with a soluble stimulus (anti-IgM, anti-CD40, anti-CD3, anti-CD28) (Suppl. Fig. 15C), suggesting that the defect in T cells was due to reduced antigen presentation by KO B cells.

The ability of WT and KO B cells to differentiate to GC-like B cells in vitro when co-culturing with 40LB cells[63] was analysed, in which antigen presentation is dispensable. After 4 days, GL7 and CD95 expression was measured by flow cytometry as a read-out of GC

**Fig. 4 | Mitochondria shapes antigen presentation by B cells. A** Left, representative flow cytometry plots showing T follicular helper cells (T$_{fh}$, CD4$^+$, CD44$^+$ CXCR5$^+$, PD1$^+$) and, right, percentage quantification in WT and *Tfam* KO 7 days p.i. See Suppl. Fig. 1A for gate strategy. **B** Representative histograms of Bcl6 protein levels on T$_{fh}$ gated on panel (**A**). Bar charts show the quantification of MFI in WT and *Tfam* KO T$_{fh}$ 7 days p.i. **C** Left, representative plots of CFSE and CTV dilutions of WT, *Tfam* KO B, and OTII CD4$^+$ T cells after 3-days co-culture. Bar charts indicate the number of divisions of each population. See Suppl. Fig. 1B for gate strategy. **D** Representative histograms of Bcl6 protein levels of co-cultured OTII CD4 T cells of panel (**C**). Bar charts show the quantification of MFI. **E** Left, flow cytometry plot of iGB cells cultured with IL-4 for 4 days gated as CD95$^+$, GL7$^+$ within B220$^+$ population. Right, bar charts indicate the percentage of iGB cells and Bcl6 protein levels of iGB of WT and *Tfam* KO. **F** Antigen internalisation was measured by FACS according to biotinylated anti-IgM (anti-IgM-biotin) on the B cell surface. Anti-IgM-biotin was detected using streptavidin (Strep) fluorescently labelled. Representative histogram (left) and bar chart (right) showing data of Strep MFI on WT and *Tfam* KO-purified B cells. **G–I** Antigen polarisation and co-localisation with lysosomes. **G** Confocal images (100x magnification) of WT and *Tfam* KO B cells stimulated with AF647-conjugated anti-IgM for 30 min. The antigen is detected as green dots. MTOC (blue) was identified with α-tubulin and polarisation was determined by the proximity of the antigen to the MTOC. LAMP1 (red) indicates lysosomes. The multi-colour bar chart stated the degree of depolarised Ag (red), intermediate polarised Ag (yellow), and polarised antigen (green). **H** The lysosome number was determined using Fiji software from images of panel **I**. The co-localisation of antigen and lysosomes was determined by the Fiji JaCOP plug-in. PI indicates Pearson's index, and M1-M2 shows Mander's coefficient. **J** Antigen degradation (Degr band) was detected using streptavidin-HRP in non-stimulated purified B cells from WT and *Tfam* KO mice after incubation with biotinylated anti-IgM for 4 h. Vinculin was used as a loading control. IgM-Heavy chain: HC IgM-light chain: LC. The bar chart indicates the fold-change degradation normalised to Vinculin. **K**, **L** Bar charts indicate lysotracker and magic red levels measured by FACS along GC maturation (FO to mat-GC) in WT and *Tfam* KO mice 7 days p.i. **M** Scatter plot illustrating the most-enriched and most-depleted lysosome-related pathways obtained through GSEA for Gene Ontology terms and Reactome pathways. The vertical axis displays the enriched pathway categories, while the horizontal axis represents the normalised enriched score (NES) according to the Kolmogorov–Smirnov GSEA statistical test. The dot size represents the range of *p*-adjusted values obtained by the Benjamini–Hochberg correction. The colour coding of the dots indicates the number of related genes within each pathway, with darker, purple dots indicating bigger sizes. In panels **4A–G**, **L**, bar charts show the quantification of one representative experiment out of three, and error bars represent mean ± SEM, Each dot represents one mouse. In panel **2H** and **I**, up to 100 cells were counted from WT and KO mice from three independent experiments. An unpaired two-tail *t*-test was conducted. For panel **4J**, bar charts show the quantification of one representative experiment out of three with at least three biological replicates; and two-way ANOVA was conducted with Tukey's multiple comparisons test. *P* values *$p < 0.05$, **$p < 0.01$ and ***$p < 0.001$. See also Suppl. Fig. 15. Source data are provided as a Source Data file.

---

differentiation. WT and KO B cells exhibited GC-like features by day 4 to the same extent (Fig. 4E), reinforcing the idea that the defect in KO B cells must be related to antigen presentation. In vivo, this defect may inhibit T$_{fh}$ differentiation and GC maturation.

Reduced antigen presentation can be due to antigen internalisation, polarisation, or degradation defects. To determine whether KO B cells exhibit flaws in internalising antigens through the BCR, ice-cold B cells were loaded with saturating amounts of biotinylated anti-IgM, used as a surrogate antigen, and washed to remove unbound antibodies. The remaining anti-IgM on the B cell surface was quantified using fluorescently labelled streptavidin after 30 min. WT and KO B cells were found to internalise IgM-BCR to the same extent (Fig. 4F).

After antigen internalisation, the antigen is targeted to a specific polarised cellular compartment where it is degraded, and peptides are loaded onto MHCII for presentation to CD4 T cells[19]. This compartment is rapidly polarised towards the microtubule-organising centre after antigen internalisation, and defects in this process have been proven to inhibit antigen presentation[38,64]. To evaluate the role of mitochondria remodelling in antigen polarisation, WT and KO cells were stimulated with fluorescently labelled anti-IgM (surrogate antigen) for 30 min before fixation and imaging using confocal microscopy. Here, WT and KO cells were found to polarise the antigen similarly (~75% of cells presented a polarised antigen; Fig. 4G).

After antigen internalisation and polarisation, the antigen is degraded in lysosome-like compartments. To evaluate this process, ice-cold cells were loaded with saturating amounts of biotinylated anti-IgM as a surrogate antigen and then washed to remove unbound antigens. Then, cells were incubated at 37 °C to trigger antigen internalisation, polarisation, and degradation. After 4 h of incubation, cells were collected and lysed. Western blot analysis probed whole-cell lysates for biotinylated anti-IgM using streptavidin horseradish peroxidase. Antigen degradation was detected as the appearance of bands between 25 and 37 kDa (Fig. 4J). While WT B cells had degraded the antigen after the 4 h incubation, there was a complete blockage of this process in KO cells (Fig. 4J). Thus, it was concluded that a lack of *Tfam* leads to antigen degradation defects.

Mitochondria can establish a functional axis with lysosomes[42]. Given that KO cells fail to degrade antigens, it was tested whether lysosome content and function are affected in KO cells using confocal microscopy. Stimulated WT or KO B cells were incubated at 37 °C for 30 min with a fluorescently labelled anti-IgM as a surrogate antigen

before quantifying antigen internalisation (Suppl. Fig. 15D), and the number of lysosomes and their colocalization with anti-IgM were measured (Fig. 4G). Cells were stained with anti-LAMP1 for lysosome detection. A slight increase in the total number of lysosomes in KO cells (Fig. 4H) was observed, but no differences in colocalization with the antigen (Fig. 4I), suggesting that defects in antigen degradation might be explained by lysosome function.

Based on these results, lysosomal function was characterised in B cells during GC maturation in WT and KO mice at day 7 p.i. by flow cytometry. LysoTracker™, a fluorescent dye for labelling and tracking acidic organelles in cells, and Magic Red®, a dye for detecting cathepsin B activity, were used. Non-stimulated (FO) KO cells presented higher LysoTracker levels in vivo than WT cells, possibly as a compensatory effect (Fig. 4K). However, while control B cells remodelled their lysosome function, increasing the activity of lysosomal enzymes upon antigen encounter (Fig. 4K, L) that reached a maximum at pre-GC, KO cells failed to do so (Fig. 4L). This inability of KO B cells to shape lysosome function after stimulation seems the most likely cause of the defects in antigen degradation. To prove this, first, the response of WT and KO B cells in co-culture with OTII CD4$^+$ T cells was assayed, but now B cells were stimulated with anti-IgM and ovalbumin peptide-coated microspheres, which do not require lysosome degradation. At day 3, *Tfam* WT and KO B cells were found to trigger similar T cell proliferation and BCL6 induction (Suppl. Fig. 15E). Second, *Tfam* WT and KO B cells were stimulated with anti-IgM and Eα peptide-coated microspheres. The internalised Eα peptide is transferred to MHCII (I-A$^b$) without the need for lysosome degradation, and presented on the cell surface, which we detected using an anti-MHCII:Ea antibody. In *Tfam* WT and KO B cells, we observed equal Eα-presentation (Suppl. Fig. 15F). Our data support that mitochondria function shapes lysosomal function.

Mitochondria dysfunction has been shown to transcriptionally modulate lysosome function[42]. RNAseq data was used to perform new GSEA using GO terms focused on lysosome function (Fig. 4M and Suppl. Fig. 15G). Results showed that several gene sets related to lysosome function are over- and under-represented in KO cells, suggesting mitochondrial dysfunction triggers a retrograde signalling in B cells to exert transcriptional control over lysosome function.

Our results show that mitochondrial function is crucial for shaping lysosome remodelling in antigen-challenged B cells and, therefore, to mediate antigen processing. Consequently, defects in mitochondria

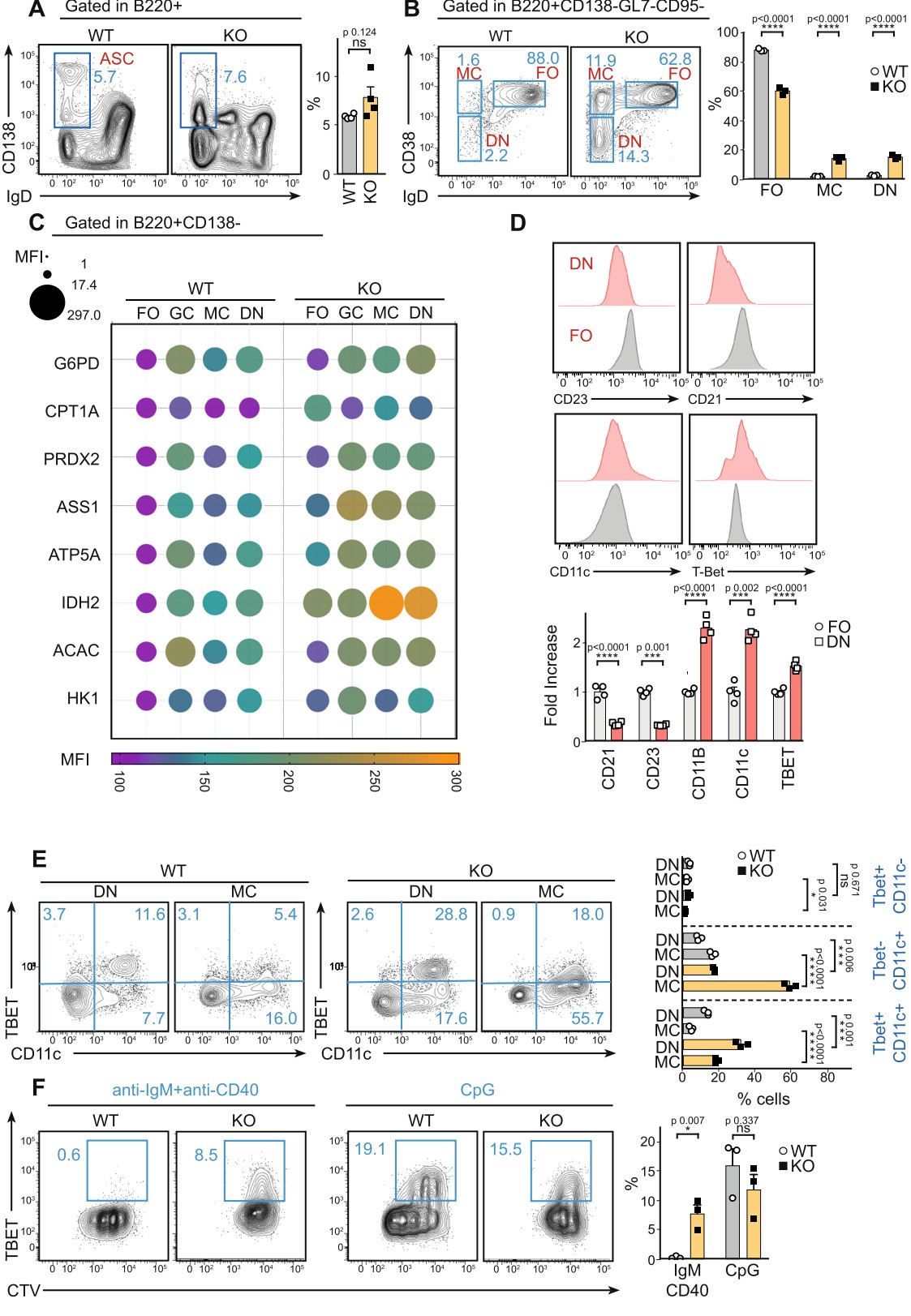

remodelling in KO B cells can lead to defects in antigen presentation to CD4 T cells, which might impact BCL6 upregulation in B and CD4 T GC cells in vivo.

## Tfam deletion leads to an aged immune response

To evaluate the effect of blockage at pre-GC on immune response outcome, the generation of antibody-secreting cells (ASCs) and MBCs

were assessed. On day 7 p.i. with SRBCs, WT and KO mice presented comparable percentages of total ASCs (IgD$^{lo}$ CD138$^+$) (Fig. 5A). However, KO mice had an elevated accumulation of IgD$^{lo}$ CD138$^{lo}$ −SLPBs (Fig. 5A and Suppl. Fig. 16A).

To evaluate the consequence of *Tfam* deletion in antibody response, WT and KO$^{Cg1}$ mice were injected with the immunogen hapten NP23-CGG in alum. NP-specific antibody titres in the serum

**Fig. 5 | Defects in mitochondria remodelling trigger ageing of the immune response. A** Flow cytometry analysis of splenic antibody-secreting cells (ASC, CD138⁺ IgD⁻) and percentage quantification of total ASC 7 days p.i. **B** Representative flow cytometry plots showing memory B cells (MC, CD38⁺ IgD⁻), FO cells (CD38⁺ IgD⁺) and double-negative cells (DN, IgD⁻ CD38⁻); bar charts indicate the percentage of FO, MC and DN cells from WT and *Tfam* KO mice 7 days after immunisation with SRBC. **C** Met-flow profile of Follicular (FO), total Germinal centre (GC), memory (MC) and double-negative (DN) B cell populations obtained by flow cytometry analysis of eight metabolic enzymes on WT and *Tfam* KO 7 days p.i. The bubble plot represents the mean value of three biological replicates. **D** Flow cytometry analysis of ABC features in DN population. Top representative histograms of CD23, CD21, CD11c and T-bet levels in WT FO and DN cells. The bottom bar chart indicates the fold increase of each value from FO. **E** Dot plot representation of Tbet and CD11c

distribution in DN and MC (panel **B**) from immunised WT and KO mice. The bar chart indicates the proportion of single-positive Tbet, single-positive CD11c and double-positive cells. **F** Dot plot representation of Tbet⁺ population in isolated B cells cultured for 3 days with either IgM/CD40 or CpG, labelled with CTV. The bar chart indicates the percentage of total Tbet⁺ cells in B cells of WT and KO mice. In panels 5A, B, D–F, bar charts show the quantification of one representative experiment out of three, and error bars represent mean ± SEM. Each dot represents one mouse. The bubble plot from panel **C** represents the mean of three biological replicates, normalised to FO WT B cell population to obtain the fold-change. In panel **A**, **D** and **F** unpaired two-tail *t*-test was conducted; for panel **B** and **E** two-way ANOVA was conducted with Tukey's multiple comparisons test:*$p < 0.05$, ***$p < 0.001$ and ****$p < 0.0001$. See also Suppl. Figs. 16–19. Source data are provided as a Source Data file.

were monitored every week for 21 days using ELISA (Suppl. Fig. 16B). Results showed that the primary IgM was comparable among WT and KO^Cg1 mice, something expected considering the Cre expression pattern of these mice. However, while WT titres of NP-specific IgG1 and IgG2b antibodies increased with time, in KO^Cg1 they were almost absent. This suggests that Tfam is fundamental in generating class-switched antibodies during the immune response (Suppl. Fig. 16B).

*Tfam* KO mice had twofold increase in the percentage of MBCs (CD38+IgD−) (Fig. 5B). MBCs can be divided into different populations according to PDL2 and CD80 expression[10,65], named single positive (SP) (CD95⁻ GL7⁻ CD38⁺ PDL2⁺ CD80⁻) and double positive (DP) (CD95⁻ GL7⁻ CD38⁺ PDL2⁺ CD80⁺) MBCs. These cells have been suggested to arise from extrafollicular and GC reactions, respectively[10,65]. On day 7 p.i., discrete SP and DP populations inside the total MBC compartment were detected in WT mice (Suppl. Fig. 16C). The overrepresented MBC compartment in KO mice was characterised by an accumulation of the SP population in the MBC compartment (Suppl. Fig. 16C). These results, together with the blockage of LLPCs (IgD^lo CD138^hi) (Fig. 5A), and defects in the production of class-switched antibodies (Suppl. Fig. 16B), suggest that activated KO B cells failing to progress into mat-GC leads to an extrafollicular response and lead to differentiation into SLPBs and MBCs.

The gating strategy to identify total MBCs showed the existence of a relevant CD38⁻IgD⁻ population in *Tfam* KO mice (Fig. 5B), henceforth termed the double-negative (DN) population. Significantly, DN was induced after SRBC immunisation and is almost undetectable in non-immunised mice (Suppl. Fig. 17A). Importantly, this DN also displayed changes in their lysosome content (Suppl. Fig. 17D).

Considering that these cells display a complete downregulation of CD38, a classical feature of GC cells[66], it appears likely that these cells are associated with the GC reaction. For more insight into the origin of DN cells, SCENITH and Met-Flow techniques were used to profile their metabolic stage. SCENITH results did not show any difference in metabolic dependencies among FO, GC, MC and DN in WT mice (Suppl. Fig. 17B, C). However, Met-Flow profiling showed that FO, GC, MC and DN presented differential metabolic profiles in WT mice, with GC being the most metabolically active pool (Fig. 5C). While MC had less metabolic activity, DN cells presented an activated profile, closer to GC than to MC. Notably, this data showed the plasticity of MC and DN B cells in their ability to adapt to lack of *Tfam* by upregulating their glucose dependency (Suppl. Fig. 17B), and also raising the rate of most metabolic pathways (Fig. 5C). This data, together with the observation that these cells displayed higher levels of activation markers CD69 and GL7 (Suppl. Fig. 17E) compared with FO, suggest that these DN cells might be GC-derived.

Indeed, DN cells were negative for CD21 and CD23 and displayed a higher forward scatter and greater levels of CD11b and CD11c compared with FO cells in WT mice (Fig. 5D and Suppl. Fig. 17F). These features are characteristic of ABCs[67–69], a pool of cells that increase with age. The ABC pool is a heterogenous population containing at least three sub-types, the most common being B220⁺Tbet⁺CD11c⁺ cells[67].

To determine whether accumulation of this DN population in *Tfam* KO mice after immunisation could be evidence of a general ageing phenomenon triggered by defective mitochondria remodelling at the pre-GC stage, the percentage of three different ABC cell pools (Tbet⁺CD11c⁻; Tbet⁺CD11c⁺; Tbet⁻CD11c⁺) in WT and KO mice were investigated after immunisation. KO mice showed a dramatic increase in the appearance of Tbet⁺CD11c⁺ and Tbet⁻CD11c⁺ABC populations (Suppl. Fig. 18A), comparable to aged mice (Suppl. Fig. 18B). These two discrete populations appeared enriched in DN and MBC populations, respectively (Fig. 5E).

To determine whether a failure in mitochondria remodelling upon antigen encounter could exacerbate the appearance of ABCs, the emergence of ABCs after B cell activation was monitored along with GC maturation. Results showed that ABC populations appeared mainly from pre-GC in WT mice (Suppl. Fig. 18D, E). *Tfam* KO mice displayed a significant increase in the percentage of ABCs appearing during GC maturation (Suppl. Fig. 18D, E).

To evaluate the intrinsic effect of mitochondria function in ABC development, in vitro experiments with WT- and KO-purified B cells were performed. The appearance of T-bet⁺ ABCs in vitro has been described to depend on innate stimuli such as TLR9[70]. Similarly, CpG stimulation pushed WT and KO B cells to the same extent towards T-bet⁺ phenotype (Fig. 5F). Notably, while stimulation of WT cells with anti-IgM plus anti-CD40 led to a poor expansion of T-bet⁺ cells in vitro, KO cells led to a tenfold increase in the induction of this population (Fig. 5F), supporting in vivo results showing the differentiation pathways of these cells upon stimulation towards ABCs.

Finally, we tested whether targeting mitochondria or lysosome function in vitro could lead to the expansion of ABC differentiation in vitro. WT cells were treated with oligomycin or pepstatin to induce mitochondria or lysosome dysfunction, respectively (Suppl. Fig. 19). In both cases, stimulated B cells increased their expansion towards ABC phenotype, supporting the link between mitochondria-lysosome function and aged immune response. Together, these results demonstrate that *Tfam* deletion, causing defective mitochondria remodelling upon antigen encounter, induces an aged immune response featuring the appearance of ABCs and MBCs.

## Discussion

Using a mouse model in which Tfam[47], a transcription factor responsible for controlling mtDNA replication and transcription, has been selectively deleted in the B cell compartment early during their development, a new role of mitochondria in B cell physiology was demonstrated. *Tfam* KO B cells could not remodel their mitochondria content after antigen encounter, resulting in an inadequate immune response. This immunodeficient phenotype is characterised by a severe blockage of activated B cells at the pre-GC stage, and appears due to a failure in lysosome remodelling upon antigen encounter. Thus, in contrast to previous studies in which mitochondria function has been linked to metabolic rewiring and bioenergetic support in proliferating B cells[36], we propose a new role for mitochondria, namely

that they are capable of shaping lysosomal content in B cells, making them indispensable for antigen presentation. Notably, a blockage at the pre-GC stage due to dysfunctional mitochondria remodelling triggers an aged immune response characterised by the accelerated appearance of ABCs.

Until now, mitochondria function in activated B cells has been linked to mTORC1 activity since mitochondria remodelling is needed to maintain the bioenergetic status of the cell and establish a specific metabolic profile after antigen encounter[46]. Failure in mTORC1 signalling has been proposed as a causal mechanism behind defects in GC reaction[36] and ASC differentiation[38,45]. *Tfam* KO mice also present flaws in the GC reaction and ASC differentiation, confirming the role of mitochondria during the GC reaction. However, these flaws are not due to mTORC1 signalling or ATP homoeostasis defects. Based on our findings, the Tfam system serves as a model to elucidate some of the ATP-independent functions of mitochondria in B cells.

Most studies that have addressed the role of mitochondria in B cells used either drugs to inhibit mitochondria function or genetically modified mice where pathways upstream of PI3K-Akt-mTORC1 were targeted[36,38,46,71,72]. Only a few studies have focused directly on B cell mitochondria[45,49,73]. A recent study by ref. 45. targeted mitochondria function in B cells using a mouse model in which B cells express a dominant-negative mutant (DNM) of the mitochondria helicase TWINKLE. Although similar to our Tfam system, the consequences of the immune response differ slightly, possibly because DNM B cells, unlike *Tfam* KO B cells, present bioenergetic stress that diminishes mTORC1 signalling. As previously shown[74], this discrepancy between DNM and Tfam systems can be explained by the fact that both models use different Cre systems, namely *CD23* and *mb1*, respectively. These Cre systems differ at the time of gene excision. Significantly, in our Tfam system, KO B cells have gone through metabolic checkpoints early during B cell development[75]. This should prompt B cells to employ compensatory effects to overcome this selection and reach optimal bioenergetic status. Still, both models reinforce the idea that mitochondria remodelling is pivotal to B cell function.

The ability of KO B cells to maintain bioenergetic homoeostasis demonstrates that B cells possess enormous metabolic plasticity to fulfil their bioenergetic requirements despite failing mitochondria remodelling. Our in vitro and in vivo results suggest that this plasticity leads KO B cells to become relatively hyper-consuming, thus internalising more glucose from the environment than their WT counterparts which increases their glycolytic capacity and dependence on glucose. The increase in FAO correlates with the upregulated expression of SDH and CPT1 observed in KO B cells. As described for other systems[58,76], this upregulation is caused by a loss of function in other electron transport chain complexes, and could explain the slightly increased levels of ATP found in FO *Tfam* KO cells. Recently, *Tfam* loss in macrophages[58] and astrocytes[59] has been shown to lead to a dramatic perturbation of lipid homoeostasis, triggering LD accumulation and a deleterious phenotype in both cases.

*Tfam* deletion in macrophages appears to have a differential impact depending on the organ of residence[58]. Our data show that changes in lipid metabolism in KO B cells does not appear to lead to an increase in LD accumulation compared with WT, suggesting that this metabolic adaptation is not the cause of the GC blockage but rather a consequence of the metabolic plasticity of B cells. However, whether the deletion of *Tfam* in B cells in other tissues could differentially affect their function, is unknown.

The GC has been described as a peculiar microenvironment with a characteristic cellular and biochemical composition, including areas deprived of oxygen[37,77]. B and T cells at the GC appear to be synchronised to some extent as they present complementary metabolic profiles[78,79]. We believe that the metabolic plasticity of KO B cells, including more glucose usage during the GC reaction, could impact the niche by forcing the different immune cells (e.g. T_fh, follicular dendritic cells) to compete for vital nutrients. Lymphocytes undergo tissue imprinting, and changes in the microenvironment can drive metabolic, transcriptomic, and proteomic changes that influence their effector functions[80–82]. Future work should examine whether the compensatory mechanisms triggered in KO B cells impact the metabolic profile of surrounding cells and, therefore, immune signalling pathways such as mTORC1, c-Myc or AMPK[77].

Metabolic profiling of GC has been challenging[60]. GC reaction and affinity maturation seems to be coupled to OxPhos and rely primarily on FAO[34]; however, some data show glycolysis also plays a critical role[37,83]. Our metabolic profiling using SCENITH and Met-Flow has shed some light on this controversy, showing that GC maturation occurs in parallel with metabolic remodelling. This suggests that different metabolic pathways, such as glycolysis and FAO, can be relevant in each stage and utilise specific roles. In fact, the transition to mat-GC correlates with diminished expression of key metabolic enzymes such as HK1 and CPT1, leading to a reduction in glycolytic and FAO capacities. One aspect that should be investigated further is the molecular pathway that drives this metabolic shunt, as this dysregulation might have pathological implications.

Our results show that mitochondria remodelling appears to shape lysosome function upon antigen encounter. These organelles and peroxisomes are essential for cellular metabolism, and their functions must be coordinated to ensure the whole cell is tuned[84–87]. It is well established that defective mitochondria lead to lysosomal impairment[88]. These vesicles become non-acidic and lose their hydrolytic activity in an ATP-mTORC1-independent fashion. Our results show that B cells lacking mitochondria remodelling after activation also fail to remodel their lysosome compartment. Recently, it has been shown that mitochondria control the translation of a specific set of proteins in cytotoxic granules of activated CD8 cells[89]. Given that a mild defect in translational levels during GC maturation was observed (Suppl. Fig. 13C), it would be interesting to examine whether this slight reduction affects lysosomal content.

It is not known how mitochondria modulate lysosome function in B cells. ROS production, physical contact, calcium uptake, protein import, or the release of vesicles, peptides, or mtDNA are possible underlying mechanisms that could be investigated[42]. RNAseq data suggests the existence of retrograde signalling in B cells, from mitochondria to the nucleus, which transcriptionally shapes lysosome function.

A failure in antigen presentation leads to reduced T_fh cell input and, therefore, defective BCL6 upregulation in B cells. This defect directly impacts GC maturation and the output of the GC reaction in that it impedes the development of a mature LLPC compartment (IgD−CD138+) but not of MBC (B220+PDL2+CD80−CD38+). This would suggest that the GC reaction is needed exclusively for ASC differentiation but not for MBC development.

BCL6 expression is known to repress CCR6, triggered during the pre-GC stage[52]. In KO B cells, less antigen presentation leads to failure of BCL6 upregulation, and consequently, CCR6 expression might not be repressed[90]. In turn, this could lead to MBC differentiation. Defects in BCL6 expression are not accompanied by induction of IRF4 (Suppl. Fig. 9F). and thus the development of ASC because IRF4 expression requires the engagement of strong, cognate T_fh cell interaction[91], which explains the failure in LLPC differentiation. These results support the idea that MBC development may be driven by weaker activation signals[12,90,92].

While it has been suggested that MBCs and LLPCs might arise from the light zone, our results contradict this assumption, at least in part. The typical boomerang shape in GC[93] with "pure" light and dark zone populations does not appear in KO mice, which are of an intermediate phenotype. This finding would support the idea of an intermediate stage in the GC[94]. It is possible that the 'pure' light zone is the precursor to ASC (absent in KOs), while the intermediate stage is the

precursor to MBC, commensurate with cells receiving low $T_{fh}$ help. Future studies should examine whether the *Tfam* KO system could be a model to explain the transcriptomic and proteomic features of such an intermediate population.

Finally, our results show that defects in B cell mitochondria remodelling may act as a trigger for an aged humoral response. Ageing is associated with intrinsic defects in CD4 T and B cells[70,95]. While mitochondria function has been demonstrated to underlie ageing in CD4 T cells[44,96,97], this concept is almost unexplored in B cells. Our results show that defects in mitochondria remodelling lead to a skewing of B cell repertoire towards the MBC compartment and the appearance of ABCs, crucial features of the ageing humoral system[70]. Crucially, our in vitro findings indicate that intrinsic deficiencies in mitochondria result in ABC differentiation. This suggests that the functional outcome may be unrelated to defects in antigen presentation, highlighting mitochondria dysfunction as the pivotal event in both scenarios.

ABCs have been described as antigen-experienced cells arising from T cell-dependent immune responses in the spleen[67,68]. These cells increase during ageing and are considered a heterogeneous pool of cells. This heterogeneity suggests different development routes dependent on environmental factors[67]. Specific ABC populations have been proposed as intermediate stages towards ASC differentiation[98]. Our results indicate a particular ABC pool, Tbet⁺CD11c⁺, originates from the GC reaction, specifically during the pre-GC stage. These cells may have a unique role or represent an intermediate stage towards another type of ABC, such as those enriched in the MC compartment (i.e. Tbet⁻CD11c⁺). Also, they could be an intermediate stage towards other functional cells, such as ASC precursors that have been recently identified[99]. To better understand ABC ontology, performing single-cell RNA sequencing analysis to trace the origin, development, and characterisation of the different ABC subsets in our Tfam system may be informative. ABCs are related to immunosenescence, autoimmune diseases[100], and cancer[101]; whether mitochondrial dysfunction in the B cell compartment underlies these processes is something to be explored.

In summary, we have identified a new mitochondrial function in B cells: they not only serve as the energy source but also control lysosomal function. Defects in mitochondria remodelling impact the humoral response, impeding GC maturation and leading to an aged response characterised by a skewed B cell repertoire towards MBC and ABC cells that might have pathological implications.

# Methods

## Mice
*Tfam*$^{fl/fl}$ mice[47] were provided by Larsson NG (Max Plack Institute for Biology of Ageing, Cologne, Germany) and were crossed at the Centro de Biologia Molecular Severo Ochoa (CBMSO) animal facility with MB1$^{Cre}$ mice[48], and Cγ1$^{Cre}$ mice[51]. MB1$^{Cre}$, Cγ1$^{Cre}$ and OT2[102] mice were kindly provided by Alarcón B (CBMSO, Madrid, Spain). Mouse colonies were bred at the CBMSO under specific pathogen-free conditions and on a C57BL/6 background. Mice were group-housed, have not been used in previous procedures and were fed with standard chow. Littermates were randomly assigned to experimental groups. Males and females between the ages of 8–16 weeks were used for all experiments. Euthanasia was performed by $CO_2$ asphyxiation followed by cervical dislocation. The ethical committee of the CBM Severo Ochoa, and Comunidad de Madrid approved all procedures. All animal procedures conformed to EU Directive 86/609/EEC and Recommendation 2007/526/EC regarding the protection of animals used for experimental and other scientific purposes, enforced in Spanish law under Real Decreto 1201/2005.

## Primary cell culture
Primary cells were obtained from the above-mentioned male and female mice, and the experimental methods are described in the method details section.

## Immunisation and ELISA
Male and female mice (equally used and randomly assigned to experimental groups) of 6- to 12-week-old mice were immunised intraperitoneally with $2 \times 10^9$ sheep red blood cells (SRBC). Spleens were harvested 7 days post-immunisation (p.i.).

For antibody detection, mice were injected intraperitoneally with NP-CGG (0.5 mg/ml) (Bio Research) in Imject Alum (0.5 mg/ml) (ThermoScientific), and blood samples collected from the sub-mandibular vein on days 0, 7, 14, 21 after immunisation. Flat-bottom 96-well immunoplate (Thermo Fisher Scientific) were coated with either NP27-BSA, NP7-BSA or biotinylated anti-mouse IgM and IgG (Southern Biotech) followed by a blocking step with Bovine serum albumin (Sigma). Absorbance at 405 nm was determined with a SPECTRAmax190 plate reader (Molecular Device).

## Cell isolation, labelling and culture
Splenic naive B or CD4⁺ T lymphocytes were purified using negative B cell or CD4⁺ T cell isolation kits (Miltenyi Biotec), yielding enriched populations of 95–98% (B cells) and 80% (T cells), respectively. Purified T or B cells were incubated for 5 min at 37 °C with 5% $CO_2$ in PBS with 2.5 µM Cell Trace Violet (CTV, Invitrogen), 2.5 µM CFSE (Invitrogen), or 5 µM Cell Trace Blue (CTB, Invitrogen). When necessary, cells were maintained in RPMI 1640 supplemented with 10% FCS, 1 mM Gluta-mine, 0.01% sodium pyruvate, 20 mM 2-mercaptoethanol, penicillin and streptomycin, from now, complete cell medium.

## B cells activation and proliferation analysis
Isolated B cells were maintained in a complete medium supplemented with 10 ng/ml of IL-4 and IL5 (Petrotech) and stimulated under different conditions: 1 or 5 µg/ml anti-IgM F(ab)2 fragment (Jackson ImmunoResearch), 0.3 µg/ml anti-CD40 (Thermo Fisher) or 3 µg/ml CpG (sigma).

## Antigen internalisation and degradation
Internalisation assays were performed as follows: purified B cells were loaded with biotinylated anti-IgM antibodies in a complete medium for 15 min on ice. Subsequently, cells were washed with a complete medium to remove unbound anti-IgM and incubated for 30 min at 37 °C. Cells were then fixed with 4% formaldehyde and incubated with streptavidin-APC (Biolegend) to detect the biotinylated anti-IgM on the cell surface by flow cytometry.

For antigen degradation, purified B cells were incubated with biotinylated anti-IgM at 37 °C for 4 h. Then, cells were lysed to detect biotinylated anti-IgM fragments by immunoblotting using streptavidin-HRP (see Immunobloting method).

## Co-culture experiments (B:CD4⁺ OTII cells)
B cells were isolated from WT, *Tfam* KO mice and CD4⁺ T cells were purified from OTII mice using negative selection kits for B and CD4⁺ cells (Miltenyi Biotec), respectively. As previously described, B cells were stained with 2.5 µM CFSE and CD4⁺ cells with CTV 2 µM. B cells were pre-incubated with biotinylated anti-IgM and biotinylated oval-bumin (OVA) or biotinylated anti-IgM and biotinylated OVA peptide 323–339 (OVAp) coated beads for 30 min at 37 °C and then washed with a complete medium. Finally, $2 \times 10^5$ B cells and $2 \times 10^5$ CD4+ cells/well were seeded in a 96 multi-well plate and cultured for 3 days. For soluble antigen stimulation experiments, B and T cells were co-cultured for 3 days in the presence of 10 ug/ml anti-IgM, 0.3 ug/ml anti-CD40, 0.3 ug/ml anti-2c11 and 0.3 ug/ml anti-CD28.

## Beads preparation
Beads used in co-cultures were prepared as follows: 0.11 mm streptavidin-coated red microspheres (Bangs Laboratories) were incubated with saturating amounts of biotinylated anti-IgM and bio-tinylated OVA or OVAp for 1 h at 37 °C. Beads were then washed and

resuspended in a complete medium. Efficient titration of the IgM signal was measured by flow cytometry.

OVA/OVAp was biotinylated in-house, using Sulfo-NHS-LC-biotina EZ-Link™ (Thermo Fisher, 21335). 0.5 mg/mL OVA were incubated with 20 µg/mL Sulfo-NHS-LC-biotina EZ-Link™ for 30 min, and free biotin was removed by dialysis.

Beads for antigen presentation with Eα were prepared as follows: 0.11 mm streptavidin-coated red- or non-fluorescent microspheres (Bangs Laboratories) were incubated with saturating amounts of bio-tinylated anti-IgM and biotinylated Eα for 1 h at 37 °C. Beads were then washed and resuspended in B cell complete medium. Efficient titration of the IgM signal was measured by flow cytometry. Biotin-GSGFAKFASFEAQGALANIAVDKA-COOH was produced by the Crick Peptide Chemistry facility.

### Eα antigen presentation

To detect antigen presentation, B cells loaded with Eα peptide and IgM-coated beads were incubated for 5 h at 37 °C, followed by fixation in 4% formaldehyde. Subsequently, these cells were stained with an anti-MHCII:Eα antibody, followed by a proper cocktail of antibodies to detect B cells and MHCII levels by flow cytometry.

### B-T conjugate assay

To determine the amount of B-T cell conjugates, B cells from WT and Tfam KO mice were first purified, along with CD4 T cells from OTII mice. OVA- or OVAp-coated beads were prepared as previously described. B cells at a concentration of 10 million cells/ml were then incubated with either OVA- or OVAp-coated beads for 30 min at 37 °C. Subsequently, the cells were washed in a complete medium and resuspended at 2 million cells/ml. B cells were co-cultured with 2 million cells/ml of CD4 T cells for 30 min at 37 °C. The cells were then fixed for 40 min with 4% PFA and proceeded to surface staining with AF488-B220 and AF647-CD4.

### Nojima Culture (in vitro induced germinal centre B cells – iGB)

The iGB culture system was described previously by ref. 63. Briefly, a 40LB cell line, a cell line derived from a 3T3 fibroblast cell line and stably expressing CD40 ligand and B cell activating factor (BAFF), was maintained in high-glucose DMEM supplemented with 10% FCS and penicillin/streptomycin either in p100 tissue culture plate (Corning, ref 353003) or multi-well six plates. Once cells reached confluent, they were detached using trypsin/EDTA treatment,

washed, counted, and collected in 15-mL tubes with 10 mL of medium before irradiation (80 Gy).

After irradiation, cells were washed, and $3 \times 10^6$ cells were seeded per p100-mm dish or $5 \times 10^5$ cells per well of a multi-well six plate. Following an overnight attachment in cell culture conditions, the 40LB medium was removed, and $5 \times 10^5$ (p100-mm dish) or $5 \times 10^4$ (multi-well six plates) and CTV-labelled naïve B cells were added on top of the feeder 40LB layer for 4 days. The germinal centre-like phenotype was analysed by flow cytometry using B220, CD95, GL7, CD38 and IgD markers.

### Flow Cytometry

For analysis of splenocyte populations, single-cell suspensions were prepared from homogenised spleens. Erythrocyte lysis was per-formed with ACK buffer (0.15 M NH4Cl, 10 mM KHCO3, 0.1 mM EDTA; pH = 7.2−7.4). Cells were treated with the appropriate combi-nation and dilution of the following antibodies: Fc block CD16/32 Fc block (1/100), B220 (1/400), CD19 (1/400), CD95 (1/400), GL7 (1/400), IgD (1/400), IgM (1/400), CD38 (1/500), PDL2 (1/200), CD138 (1/300), CD86 (1/100), CXCR4 (1/100), CD69 (1/400), CCR6 (1/200) c-Kit (CD117) (1/200), CD23 (1/200), CD21 (1/200), CD5 (1/200), CD11b (1/200), CD11c (1/200), CD4 (1/400), CD44 (1/400), PD1 (1/100), CXCR5 (1/100). Dead cells were excluded using fixable viability markers

ghost-dye red 780 or ghost-dye violet 540 (TONBO Biosciences) (1/2000). For intracellular detection of Bcl6 (1/300), T-bet (1/100), Ki67 (1/1000), cleaved-Caspase3 (1/500), pS6 (1/500), pAkt (1/500) or Perk (1/500), cells were fixed and permeabilised either with eBioscience Foxp3/Transcription Factor Staining Buffer Set Invitrogen, #00-5523-00; for nuclear detection of Tbet, Bcl6 or Ki67) or BD Cytofix/Cyto-perm™ Fixation/Permeabilization Kit (Cat No 554717); for cytoplasm detection of cleaved Caspase 3 and p-S6, pAkt or pERK. Intracellular primary antibodies cleaved-Caspase3, pS6, pAkt or pERK were dilu-ted in the appropriate 1x Perm/Wash buffer. Alexa488 donkey-anti-Rabbit IgG antibody (Life Technologies) was used as a secondary antibody to detect Cleaved caspase 3, pS6, pAkt or pErk. The anti-body was diluted in 1xPerm/wash buffer and incubated for 40 min at room temperature. Mitochondrial status and cellular ROS were measured with MitoTracker Green (20 nM) and MitoTracker Red CMXRos (50 nM) or CM-H2DCFDA (2.5 mM), respectively. All tracers were incubated in supplemented-free RPMI for 30 min at 37 °C. For substrate influx assays, up to $2 \times 10^6$ single-cell suspensions were suspended in B cells complemented and incubated for 30 min at 37 °C with 25 ug/ml of 2-NBDG (Abcam) (glucose uptake) and 500 nM BODIPY™ FL C12 (Thermo Fisher) (long chain fatty acid uptake). Cells were then pelleted and washed once in PBS 2% FBS. Lipid droplet content was detected by incubating cells for 15 min at room tem-perature with 1ug/ml of Nile Red diluted in PBS. Acidic lysosome content was determined by incubating cells for 30 min at 37 °C with 50 nM Lysotracker Green DND-26 in B complete RPMI. Cathepsin activity of lysosomes was measured by incubation with a Magic Red probe for 20 min at 37 °C in RPMI. Stock solution (25 test) was reconstituted in 50 ul of DMSO, then an intermediated dilution was prepared (1 ul of stock in 360 ul of PBS), and the final concentration of the probe was 1:10 in complete RPMI. As described previously, ATP content was determined with fluorescent probe 5 µM quinacrine[103]. Briefly, single-cell suspension (range $5 \times 10^5$–$10 \times 10^6$ cells/ml) was incubated in 5 µM of Quinacrine diluted in KREBS medium (glucose-free medium). Data were acquired on FACSCanto II Flow Cytometer or Cytek Aurora full spectrum cytometer and analysed with FlowJo software.

### Met-flow cytometry staining

Based on ref. 61, eight metabolic proteins were chosen and optimised for flow cytometry analysis of murine lymphocytes. Purified metabolic antibodies ASS1 (1/150 once conjugated), G6PD (1/300 once con-jugated), HK1 (1/200 once conjugated) and ACAC (1/500 once con-jugated) were purchased and conjugated with Dylight 680 (ab201804), Lightning-link PeCy7 (ab102903), Dylight 405 (ab201798) and Lightning-link APCCy7 (ab102859) respectively. Antibodies were pur-chased pre-conjugated for other enzymes: CPT1A-AF488 (1/200), IDH2-PE (1/250), PRDX2-AF647 (1/150) and ATP5A-AF594 (1/150). In the final met-flow panel, metabolic enzymes were combined with Ghost-dye540 and lymphocyte markers such as CD4, CD8, IgD, CD11c, Tbet, CD38, GL7, CD95 and B220 in the above-mentioned dilutions. Splenic lymphocytes were obtained as previously described. Extracellular staining was performed at 4 °C for 20 min using Brilliant Stain Buffer (BD, 563794) diluted in PBS 2% FBS. Following incubation, cells were washed, fixed and permeabilised using eBioscience Foxp3/Transcrip-tion Factor Staining Buffer Set according to the manufacturer's instructions.

The cells were then washed in the provided wash buffer and blocked in wash buffer supplemented with 20% FBS for 30 min at room temperature. Finally, cells were stained with intracellular anti-bodies in a permeabilisation buffer for 2 h at room temperature. Subsequently, cells were washed once in permeabilisation buffer, followed by a PBS 2% FBS wash. Samples were acquired on a Cytek Aurora 5 L spectral cytometry instrument, and analysis was per-formed using FlowJo.

## SCENITH

For the SCENITH assay, cells were seeded at $10^7$ cells per ml in 96 round bottom well plates and treated for 45 min with control, 2-Deoxy-D-Glucose (DG, 100 mM), Oligomycin (Oligo, 1 μM), Etomoxir (10 μM) and CB-839 (1 μM) or a combination of these drugs. Puromycin (10 μg/mL) was added with the metabolic inhibitors. Cells were washed in cold PBS and stained with a combination of Fc receptors blockade and fluorescent cell viability marker. Subsequently, cells were stained with primary conjugated antibodies against surface markers for 20 min at 4 °C in PBS 2% FBS. Cells were then stained intracellularly for puromycin following Foxp3/Transcription factor staining kit (eBioscience), and AF488-conjugated anti-puromycin monoclonal antibody for 1 h at 4 °C[60]. Cell dependencies and capacities were obtained as previously described[60].

## ATP luminescence assay

For ATP quantification by luminescence, the ATP Determination kit (Thermo Fisher, A22066) was used following the manufacturer's instructions. Briefly, $1.5 \times 10^5$ non-stimulated or IgM + CD40-stimulated purified WT or *Tfam* KO B cells were lysed after 24 h culture with Tris-EDTA buffer (Sigma, 93302) for 15 min at 4 °C. Lysates were centrifuged at maximum velocity for 10 min at 4 °C. The standard reaction solution was prepared as specified, and the standard curve and sample measurements were obtained in CLARIOstarPlus (BMG LABTECH).

## Confocal microscopy and image analysis

About $10^5$ purified B cells were seeded in poly-L-lysine treated IBIDIS plates and incubated for 15 min at 37 °C 5% $CO_2$ for cell adhesion. Cells were then stimulated with 10 μg/mL AF647-conjugated anti-IgM (Jackson ImmunoResearch) for 30 min at 37 °C 5% $CO_2$. Cells were then fixed with 4% formaldehyde, permeabilised with 0.01% saponin 0.5% BSA and blocked with 0.01% saponin 3% BSA. Staining was performed in a permeabilisation buffer with the following antibodies: anti-tubulin, anti-LAMP1, AF488-conjugated anti-mouse and AF555-conjugated anti-rat. Confocal imaging was performed with an LSM800 microscope (Carl Zeiss) with a 100X objective.

For spleen immunofluorescence, spleens of immunised mice were fixed in 4% PFA and dehydrated with a sucrose gradient overnight. Then, they were embedded in OCT and stored at −80 °C. Frozen sections that were 15 μm wide were cut using a cryostat. Sections were hydrated with PBS and incubated with 10 mM ammonium chloride for 10 min at room temperature. After two washes with PBS, sections were blocked for 1 h with PBS 2%BSA 5% FBS. Surface antibodies were incubated in a blocking buffer with 0.05% Triton X-100 overnight: B220-647, GL7-AF488 and CD3-PE. For image analysis, Fiji (ImageJ) software was used. Co-localisation data were obtained using the JACoP plug-in.

## Immunoblotting

For detection of antigen degradation and Tfam by western blot, purified B cells were lysed in lysis buffer containing 20 mM Tris-HCL, pH 8.0, 150 mM NaCl, 5 mM EDTA, protease inhibitor cocktail (Santa Cruz Biotechnology), 10 mM NaF, 1 mM Na3VO4, and 1% NP40 for 30 min on ice. Samples were loaded into 10% SDS-PAGE gel (BIO-RAD). Proteins were detected with streptavidin-HRP and primary antibodies against Vinculin, Tfam, and USP7 using the secondary HRP-conjugated anti-rabbit or anti-mouse antibodies (for more information, see Key Resources Table).

## Electron microscopy

The Electron Microscopy Facility prepared a sample at the CBMSO, UAM University, Spain. Cell samples were fixed in 2% glutaraldehyde and 4% formaldehyde in 0.1 M Phosphate buffer pH 7.4 overnight. Overnight fixation was followed by buffer washes and post-fix treatment with 1% osmium tetroxide and 0.8% aqueous potassium ferrocyanide solution for 1 h at 4 °C. Samples were rinsed in distilled $H_2O$

and further treated with 0.15% tannic acid for 1 min at room temperature, followed by buffer and distilled $H_2O$ washes and uranyl acetate 2% treatment for 1 h. Cells were embedded in 10% gelatin and further dehydrated through a graded acetone series. Cells were included in TAAB 812 resin (TAAB laboratories, Berkshire, England). Ultra-thin sections (70 nm) were then mounted on copper grids and stained with 2% uranyl acetate in water for 7 min and with Reynolds lead citrate for 2 min. Samples were examined on a Jeol JEM-1010 electron microscope (Jeol, Japan) 4,000× and 15,000× objectives. Mitochondrial classification and mitochondrial area were segmented manually and analysed using Fiji (http://fiji.sc/Fiji) and ImageJ 1.48 v software. The investigator was blinded to the group allocation when assessing the outcome.

## Blue Native page electrophoresis and western blotting

About $30 \times 10^6$ B cells were permeabilised in 200 μl of PBS and 200 μl of 8 mg/ml Digitonin at 4 °C for 10 min, centrifuged, and pellets were suspended in 80 μl of 1.5 M of aminocaproic acid, 50 mM Bis-Tris/HCl pH 7 and solubilised with 10 μl of dodecyl maltoside (1.25%) for 5 min at 4 °C and centrifuge. The supernatant was collected, and 5% Serva Blue G dye in 1 M6-aminohexanoic acid was added to one-third of the final sample volume. Equal amounts (20 μg) of samples were separated by 3–13% gradient Blue Native gel[104]. After electrophoresis, gels were electroblotted onto Hybond-P-polyvinylidene fluoride (PVDF) membranes (GE Healthcare) and immunoblotted with specific antibodies for complex I (anti-NDUFA9, Abcam ab14713), complex IV (anti-COI, Invitrogen MTCO1 459600), complex III (anti-UQCR2, Proteintech), complex II (anti-SDHA Invitrogen 459200) and complex V (homemade anti-mouse IF1, kindly gift of Jose Manuel Cuezva's laboratory, CBMSO-CSIC-UAM, Spain).

## Bulk RNA sequencing and analysis

Purified B cells were stimulated for 24 h, as previously described. Total RNA was extracted using QIAGEN RNeasy Mini KIT. Messenger RNA was purified from total RNA using poly-T oligo-attached magnetic beads. The library was checked with Qubit and real-time PCR for quantification and a bioanalyser for size distribution detection. Quantified libraries were pooled and sequenced by paired-end read on the Illumina platform. Raw data on fastq format were processed to remove reads containing adaptor poly-N and low-quality sequences. All the downstream analyses were based on clean data with high quality. The index of the reference genome was built using Hisat2 v2.0.5, paired-end clean reads were aligned to the reference genome using Hisat2 v2.0.5. We selected Hisat2 as the mapping tool because it can generate a database of splice junctions based on the gene model annotation file and, thus, a better mapping result than other non-splice mapping tools. FeatureCounts v1.5.0-p3 was used to count the reads numbers mapped to each gene. FPKM of each gene was calculated based on the length of the gene and the reads count mapped to this gene. Downstream analysis was performed in the R environment (version 4.2.2) using the Bioconductor package DESeq2 (V1.38.2) for gene count normalisation (size factor normalisation method). For differential gene expression analysis, the "apeglm" method was used to obtain unbiased logFC estimates, specifically targeting the accurate identification of significant differences in the dataset. Genes with adjusted $p$ values <0.05 after the Benjamini−Hochberg correction were identified as differentially expressed genes (DEGs). Fast Gene Set Enrichment Analysis was performed with the fgsea package (V1.2.4.0) in an R environment. Ranked gene lists derived from DESeq2 test statistics were compared to predefined gene sets. The MitoCarta 3.0 database[57] and the mouse gene sets from MSigDB[105] of Gene Ontology, Hallmarks and Reactome with 1000 permutations were utilised. To show sample clustering, heatmaps of the top DEGs in lysosome-related pathways were created using the ComplexHeatmap package (V2.14.0) from the Bioconductor environment.

## Metabolomic analysis

Cell homogenates were prepared by adding 1:1 v/v water: methanol and sonicated with a dr.hielscher UP200S (dr.hielscher, Berlin, Germany), set up to 80% intensity, 0.5 s/pulse, 16 pulses. For GC-MS, 100 µL of each homogenate was mixed with 300 µL of cold methanol containing 25 ppm D-palmitic acid (internal standard), vortex mixed for 60 min and centrifuged at 4000×$g$ for 20 min. About 250 µL of the supernatant were transferred to a vial with a glass insert and evaporated to dryness in a Gyrozen HyperVac VC2124 (Gimpo, Korea) coupled to an LVS 110Z (Gardner Denver Thomas GmbH Welch Vacuum, Fürstenfeldbruck, Germany). The samples were then submitted to methoximation (with O-methoxyamine) for 16 h and silylation for 1 h at 70 °C with $N,O$-Bis(trime5j,thylsilyl)trifluoroacetamide (BSTFA) and trimethylchlorosilane (TCMS), and finally resuspended in 100 µL heptane. The samples were injected onto a DB5-MS column (30 m × 0.250 mm, 0.25 µm) with a 10 m pre-column (Agilent Technologies, Santa Clara, CA, USA). Helium was used as a mobile phase at a constant flow rate (1 mL/min) in a GC-Q-MS (8890 GC coupled to 5977B MS) system (Agilent Technologies) with an Electron Ionisation (EI) source at 70 eV.

## Extracellular flux assay (SEAHORSE)

Naive and activated B cells were suspended in Seahorse medium (pH 7.4) supplemented with 11 mM glucose and 2 mM pyruvate. Cells were settled on a 96-well assay plate (Seahorse Bioscience) coated with Cell-TAK (Corning). OCR was recorded with the XF96 Extracellular Flux analyser. Seahorse XF Cell MitoStress kit and Seahorse XF Real-Time ATP Rate Assay kit were performed following the manufacturer's instructions.

The mitochondrial FA metabolic test was conducted using the Seahorse XF FA Oxidation Test Kit. Purified B cells unstimulated or stimulated for 24 h were suspended in Seahorse RPMI medium supplemented with 2 mM Glucose and 0.5 mM L-carnitine. BSA or palmitate-BSA was added to each cell well before the assay initiation. To determine CPT1-mediated FA catabolism, etomoxir (10 µM) was injected in port A, followed by subsequent oligomycin, FCCP and rotenone/antimycin A injections.

## Reverse transcription-quantitative polymerase chain reaction

Splenic B cells, CD4$^+$ and CD8$^+$ T cells from WT and *Tfam* KO mice were sorted in a BD FACAria Fusion. Total RNA was extracted from sorted or purified B cells using an RNeasy kit (QIAGEN) according to the manufacturer's instructions. RNA was reverse-transcribed using a High-Capacity RNA-to-cDNA Kit (Applied Biosystems), and the reverse-transcribed RNA was amplified with the appropriate primers (listed in Key Resources Table) using GoTaq qPCR Master Mix (PROMEGA). All primers were designed to hybrid two joined exons and span at least one intron. The abundances of analysed genes were performed in CFX 384 (Bio-Rad). Each value was normalised by the β-Actin gene and expressed as the relative RNA abundance to that of cells obtained from WT mice.

## Measurement of mtDNA copy number

mtDNA copy number was assessed by qPCR analysis of total DNA samples extracted from purified B cells. mtDNA content was quantified according to Quiros et al. 2017. A comparison of ND1 expression relative to Hexokinase 2 (HK2) DNA expression measured the mtDNA copy number to genomic-DNA copy number ratio.

## High dimensional analysis and clustering

Multidimensional analysis and dimension reduction were performed using UMAP implementation in OMIQ.ai (Santa Clara, CA, USA). The markers used for UMAP calculations are indicated in Fig. 4 and Suppl. Fig. 6. Up to 3000 manually gated B220$^+$ GL7$^+$ CD95$^+$ (referred to as

GCt) events were downsampled and subjected to dimensional reduction. Clusters were generated using the FlowSOM algorithm.

## Quantification and statistical analysis

Statistical analyses were performed using GraphPad Prism 8.0 software. Results are presented as mean ± SEM. Error bars and statistical tests used in each experiment and the adjustments for multiple comparisons are specified in the figure legend. Biological replicates are displayed as individual data points, and their number ($n$) is specified in figure legends. *$P < 0.05$, **$P < 0.01$, ***$P < 0.001$, ****$P < 0.0001$, ns if $P > 0.05$.

## Reporting summary

Further information on research design is available in the Nature Portfolio Reporting Summary linked to this article.

## Data availability

The bulk cell RNA sequencing data generated in this study have been deposited in the Gene Expression Omnibus (GEO) database under the accession number GSE249893. Metabolomic data generated in this study have been deposited in the Metabolomics Workbench: An international repository for metabolomics data and metadata, metabolite standards, protocols, tutorials and training, and analysis tools (2016)'. [PubMed: https://www.ncbi.nlm.nih.gov/pubmed/26467476/]. This study is available at the NIH Common Fund's National Metabolomics Data Repository (NMDR) website, the Metabolomics Workbench, https://www.metabolomicsworkbench.org where it has been assigned Study ID ST003047. The data can be accessed directly via its Project https://doi.org/10.21228/M8HH8R. This work is supported by NIH grant U2C-DK119886. The study is available for review at http://dev.metabolomicsworkbench.org:22222/data/DRCCMetadata.php?Mode=Study&StudyID=ST003047&Access=ThgJ8001. Source data are provided with this paper. Any additional information required to reanalyse the data reported in this paper is available from the lead contact upon request. Source data are provided with this paper.

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

## Acknowledgements

The research is supported by grants: HR22-00447 (NMM) from ¨La Caixa¨. Also, RTI2018-101586-A-I00 (N.M.-M.), PID2021-126298OB-I00 (N.M.-M.), PID2020-114054RA-I00 (S.C.), PID2021-122490NB-I00 (FRS), and PID2020-120367GB-I00 (F.M.-B.), all financed by the Ministerio de Ciencia, Innovación y Universidades. M.I.-P. is supported by PRE2019-089076, and J.G.-C. by PRE2021-097661, both from the Ministerio de Ciencia, Innovación y Universidades, J.R.G. is supported by PEJ-2020-AI/BMD-19546 and M.V.d.l.E. by PEJ-2019-TL/BMD-13708, both from Consejeria de Ciencia, Universidades e innovación Comunidad de Madrid. C.C. is supported by P2022/BMD7209 (INTEGRAMUNE-CM) from the Comunidad de Madrid. S.C. is supported by RYC-2017-23013 and N.M.-M. by RYC-2016-20173, financed by the Ministerio de Ciencia, Innovación y Universidades. Editorial assistance, in the form of language editing and correction, was provided by XpertScientific Editing and Consulting Services. We are grateful to the Flow cytometry and Animal facilities at the CBMSO. Finally, we are incredibly thankful to Balbino Alarcón, Ana Martínez-Riaño, Mauro Gaya, Ana Ortega and Almudena Ramiro for their critical manuscript reading.

## Author contributions

Methodology, investigation, analysis, visualisation and validation, M.I.-P., J.R.G., M.V.d.l.E., B.S.E., E.R.B. and N.M.-M.; maintained and supervised mouse colony management, C.P.C., T.G.M. and M.V.d.l.E.; performed electron microscopy experiments, J.G.-C., S.C. and M.G.R.; performed metabolomic analysis, M.F.R.-S. and F.J.R.; performed bioinformatic analysis, C.C.; performed blue native polyacrylamide gel electrophoresis, J.G.-C. and S.C.; essential reagents and support, R.J.A. and F.M.-B.; writing of original draft, N.M.-M.; funding acquisition, supervision and project administration, N.M.-M.; editing of draft, S.C., F.M.-B., E.R.B. and N.M.-M. All authors gave feedback on the paper.

## Competing interests

The authors declare no competing interests. R.J.A. discloses a potential conflict of interest as the developer and patent holder of SCENITHTM and clone R4743L-E8 adapted for SCENITH. Plans are underway to license these to GammaOmics, a company yet to be established, of which R.J.A. will be a partner. Currently, no commercial license exists, and the company is not formally established.
