## [Peer Review File · Nature Communications]

Defective mitochondria remodelling in B cells leads to an aged immune responseREVIEWER COMMENTS

Reviewer #1 (Immunometabolism) (Remarks to the Author):

In this manuscript, Iborra-Pernichi et al explore the in vivo role of mitochondrial dysfunction in B cells, which was addressed by a B-cell specific Tfam knock out. The authors show that mitochondrial remodeling and oxidative phosphorylation is essential for pre-GCs to transition into mature GC. Deficits herein lead to an accumulation of B-cells resembling key features of age-related B cells.

Major points:

1) As a key regulator of mitochondrial DNA, Tfam deletion results in the depletion of mtDNA and induces a deficiency in OXPHOS, thereby strongly impairing cellular energy metabolism and ATP production. Recently, the central role for Tfam and proper mitochondrial OXPHOS in the energy metabolism of tissue resident macrophages and astrocytes has been published (PMID: 36959514, PMID: 36738738). Many effects of Tfam B cell-KOs are similar eg. TCA deregulation, increased compensatory glucose usage and glycolysis etc. However, some more read-outs would help to get the full metabolic picture and would be needed to compare the effects between B cells and the other published cell types. i.e. The complete degradation of fatty acids requires functional OXPHOS to accept electrons from FAO and TCA generated reducing equivalents NADH and FADH₂, thus OXPHOS deficiency can result in aberrant lipid accumulation due to limited FAO. Increased C12-Bodipy uptake and elevated FAO dependency (with etomoxir) are indicative of similar effects and it would be interesting to see if Tfam KO B cells also show perturbed lipid content.

Importantly (line 292, 294 and 305), the observed effect between pre-GC and mature-GC is not a switch-off of glycolysis! Glucose dependence is still very high in ma-GC, what is observed is the reduction in glycolytic capacity, meaning how efficiently cells can switch to glucose usage if other metabolic pathways are inhibited.

How can the authors explain the discrepancy between the estimated dependencies of B cell subsets on FAO/AOO obtained by Scenith vs. FAO dependency by Etomoxir inhibition? It seems to be contradictory especially for WT vs. KO Mat-GCs

2) The link to lysosomal function and antigen processing is interesting but, in some parts, lacks mechanistic insight. Especially antigen degradation in lysosomes seems to be perturbed upon Tfam KO. Additional information about lysosomal acidification, protease content/activity would give more detailed information. Additionally, I don't really understand the point of the Fold change representation and even find it partially misleading. It seems that in vitro B-cells already exhibit the same lysosomal content as stimulated WT B cells. Similarly in the in vivo subsets it seems as if there is already very high lysosomal contents throughout the different stages. Therefore, in my opinion, it is not really correct to say that KO B cells fail to remodel lysosomes upon antigen encounter. Maybe lysosomes are dysfunctional and therefore the increases in lysosomal content is a compensatory mechanism.

3) The finding of the age-associated B cell population in conditional Tfam KO animals upon antigen challenge is very interesting. It would be very informative to also look at bioenergetic characteristics, mitochondrial and lysosomal parameters of these ageing associated population.

Minor points:

1) Does the spleen architecture in Tfam B cell -KOs change?

- 2) Can the authors at least speculate why KO B cells show an increase in mitochondrial membrane potential??
- 3) What is the Etomoxir concentrations used? 10 μ M?
- 4) Quinacrine is not a very precise ATP measurement
- 5) In the T cell/ B cell coculture experiment -> I don't get the experiment with the soluble stimuli and the conclusion thereafter. As soon as CD3/CD28 is added T cells get activated just by that, no matter if B cells are there and it is completely independent of antigen-presentation by the B cells. Wouldn't a OVA-teramere stain be more relevant for checking antigen-presentation?
- 6) In line 275 authors state that B cells did not show any relevant differences in mTORC1 signaling, However in Fig 3B mature GC show a significant increase in p-S6 levels indicating a perturbed bioenergetic state.

Reviewer #2 (GC responses) (Remarks to the Author):

The germinal center (GC) response is key for establishing long-lived humoral immune responses. B cells in GC are highly proliferative and thus metabolically very active. Better insight into the metabolic demand of GC B cells is required, and in particular what the role of mitochondria in this process is. This study reports the generation and analysis of B-cell specific Tfam KO mice. Tfam is a transcription factor for mitochondria biogenesis. Tfam KO mice displayed a blockage of GC reaction, featured by increased differentiation towards memory B cells and aged related B cells, hallmarks of an aged immune response. Unexpectedly, GC blockage in Tfam KO mice did not cause defects in the bioenergetic supply. In contrast, the Tfam KO GC phenotype showed a defect in remodeling the lysosome compartment of B cells.

Overall, I found the paper very interesting, well-written and transparent. The main point is that the conclusion that the GC phenotype in vivo is caused by mitochondrial remodeling is weak/circumstantial. As Tfam plays a crucial role in bone marrow development in the pro-B cell stage, where metabolic checkpoints have been described previously [ref 49], the B cell-specific Tfam KO mice display a severe reduction in the number of splenic B cells. As a consequence, GC formation is expected to be defective and is not necessarily caused by defective mitochondria (explained below).

Major points:

- Immunization resulted in an impaired GC reaction, which the authors attribute to a failure in mitochondrial remodeling (Fig. 2). Since the KO mice already have very low B cell numbers to start with (Fig. 1A), it is expected that they will have impaired GC formation after immunization. Can the authors exclude that the defective GC response is just caused by the low B cell numbers in the KO mice (due to the block in B cell development in the bone marrow)? In other words, what is the evidence that defective mitochondrial remodeling is responsible for the GC phenotype in vivo?
- The same holds true for the data presented in Fig 5. It is clear that Tfam KO mice generate more age-associated B cells upon immunization, but what is the evidence that this is caused by defective mitochondrial remodeling (as the authors conclude in line 421)?
- What are effects of Tfam KO on circulating antibody levels (naïve vs immunization)?

- The paper would be strengthened if some of the (in vitro) findings were confirmed in human B cells (or human B cell lines)

- Did the authors attempt to rescue the Tfam KO phenotype in vitro by reintroducing Tfam expression in the cells? (to exclude off-target effects of the Cre-induced deletion)

- The results shown in Fig 4 are striking and support impaired antigen presentation by Tfam KO B cells. The 'remodeling' of lysosomes is presented as the most likely mechanism responsible for the defects in antigen degradation (line 370). However, there are other possibilities to explain the defect in antigen degradation. Is the pH (drop) and expression of cathepsins the same in KO and WT lysosomes? In addition, is the B-T conjugate formation comparable between Tfam KO and WT cells?

Minor points:

- Figure 1: there is a dramatic decrease in total number of B220+ cells in Tfam KO spleens, yet the frequencies of FO and MZ cells is comparable to WT spleens. Which subset of B cells is then responsible for the decreased B220+ population (transitional B cells, other?)

- Please also include total cell numbers of the B cell subsets in WT and KO spleens (Fig. 1A) for clarity.

- Fig 1G: OCR is impaired in KO cells after stimulation, what about the glycolysis pathway? This can be done by Seahorse ECAR analysis of WT and KO B cells to support the data in Fig. 3.

- Please check English grammar throughout the manuscript.

Reviewer #3 (Immune ageing, systems immunology) (Remarks to the Author):

Pernichi and colleagues set out to investigate the contribution of the metabolic activity in B cells to the germinal center reaction. The experimental system using mice with Tfam-deficient B cells and the experimental design are appropriate and the data are generally convincing. The authors convincingly show that mitochondrial activity in B cells is defunct and germinal center activation is impaired. Surprisingly, the B cells have sufficient metabolic plasticity to compensate for the mitochondrial defects. The authors then convincingly show that the major defect lies in a failure of the tfam-deficient to stimulate Tfh cells.

Comments

1. Given the huge difference between oxidative phosphorylation and glycolysis in ATP production, Fig. 3a and 3b are surprising. How do the authors explain the greater ATP content in KO mice early on and the increased mTORC1 activity at the mature stage.
2. The data on failed Tfh activation are convincing, however, the evidence for defective antigen presentation is only indirect. Do B cells express MHC class II and costimulatory molecules? Are antigens that do not require processing such as an ova peptide or a superantigen able to stimulate Tfh cells?

3. The observation of the generation of ABC cells is of interest, but it remains unclear how this is related to defective T cell help or mitochondrial dysfunction.

Point by Point reply to reviewers

Reviewer #1 (Immunometabolism):

In this manuscript, Iborra-Pernichi et al explore the in vivo role of mitochondrial dysfunction in B cells, which was addressed by a B-cell specific Tfam knock out. The authors show that mitochondrial remodeling and oxidative phosphorylation is essential for pre-GCs to transition into mature GC. Deficits herein lead to an accumulation of B-cells resembling key features of age-related B cells.

We thank R1 for her/his encouraging comments. We have addressed R1's feedback below.

Specific R1 comments:

Major points:

1.- R1 stated: "As a key regulator of mitochondrial DNA, Tfam deletion results in the depletion of mtDNA and induces a deficiency in OXPHOS, thereby strongly impairing cellular energy metabolism and ATP production. Recently, the central role for Tfam and proper mitochondrial OXPHOS in the energy metabolism of tissue resident macrophages and astrocytes has been published (PMID: 36959514, PMID: 36738738). Many effects of Tfam B cell-KOs are similar eg. TCA deregulation, increased compensatory glucose usage and glycolysis etc. However, some more read-outs would help to get the full metabolic picture and would be needed to compare the effects between B cells and the other published cell types. i.e. The complete degradation of fatty acids requires functional OXPHOS to accept electrons from FAO and TCA generated reducing equivalents NADH and FADH₂, thus OXPHOS deficiency can result in aberrant lipid accumulation due to limited FAO. Increased C12-Bodipy uptake and elevated FAO dependency (with etomoxir) are indicative of similar effects and it would be interesting to see if Tfam KO B cells also show perturbed lipid content".

We express our gratitude to R1 for their invaluable suggestions, which have motivated us to incorporate a new set of data into the manuscript. This addition significantly enhances the characterization of the impact of Tfam deletion in B cells and facilitates a comparative analysis with the effects of Tfam deletion in other cell types, such as macrophages.

To address the reviewer's recommendations, we adopted the following strategy:

- Conducted an RNAseq analysis comparing in vitro stimulated WT and KO B cells. This analysis, in conjunction with a Gene Set Enrichment Analysis (GSEA) utilizing published transcriptomic data of Tfam KO macrophages, revealed striking similarities in the transcriptome alterations of both cell types, particularly in metabolic pathways like fatty acid metabolism (Fig. 3B, C).
- Employed a novel technique, Met-flow, to metabolically profile B cells in vivo and in vitro. This method offered a comprehensive understanding of the metabolic remodeling occurring in B cells during germinal center maturation (Fig. 3F), showcasing significant metabolic plasticity in Tfam-deficient B cells. Notably, alterations in lipid metabolism, analogous to those observed in macrophages and astrocytes, were unveiled
- Explored changes in lipid content in B cells in vivo and in vitro using various probes. Additionally, we assessed the oxygen consumption rate (OCR) of Tfam KO relative to WT B cells in the presence of exogenous fatty acids (FA). We further added etomoxir to evaluate FA-induced OCR and endogenous FA catabolism, providing insights into the intricate interplay of lipid metabolism in these cells (Supp. Fig 13).

We extend our sincere appreciation to R1 for their insightful suggestions, as the implemented approaches in the revised manuscript now offer pertinent information regarding B cell metabolism

during the germinal centre reaction. Our findings not only underscore the similarities between the impact of *Tfam* deletion in B cells and macrophages/astrocytes but also emphasize the unique characteristics of each cell type. Notably, our results suggest that, in this instance, lipid accumulation may not be the primary driver of the observed functional consequences in *Tfam* KO B cells.

2.- R1 stated: *“Importantly (line 292, 294 and 305), the observed effect between pre-GC and mature-GC is not a switch-off of glycolysis! Glucose dependence is still very high in ma-GC, what is observed is the reduction in glycolytic capacity, meaning how efficiently cells can switch to glucose usage if other metabolic pathways are inhibited”*

We concur with the observations made by R1, and in response, we have modified the description of this phenomenon in the Results section. Furthermore, we have addressed and discussed this aspect in detail towards the conclusion of the manuscript. The additional insights gained from the Met-Flow data contribute significantly to a more nuanced understanding of the metabolic shift occurring from pre-GC to mat-GC (Fig. 3F).

In alignment with R1's insights, it becomes apparent that the transition from pre-GC to mat-GC correlates with a discernible reduction in the metabolic capacity of B cells to switch between various metabolic sources. This implies a diminished metabolic plasticity during this transition phase.

3.- R1 stated: *“How can the authors explain the discrepancy between the estimated dependencies of B cell subsets on FAO/AOO obtained by Scenith vs. FAO dependency by Etomoxir inhibition? It seems to be contradictory especially for WT vs. KO Mat-GCs.”*

We agree with the observations made by R1 regarding the notable disparity between the FAO/AOO capacity and FAO dependency, particularly evident in the mat-GC stage. In this phase, KO cells exhibit an inclination towards increased FAO dependency despite a significant decline in FAO/AAO capacity. It's crucial to recognize the distinction between these two concepts – one reflects the extent to which cells rely on a specific pathway, while the other gauges the cell's ability to adapt its metabolism to alternative substrates, such as fatty acids (FA) and amino acids (AA), in the absence of glucose.

In essence, this highlights the plasticity of cells in adapting to other substrates when glucose utilization is impeded. Our understanding of this phenomenon aligns with the recent metabolic profiling conducted using Met-flow, revealing that the mat-GC stage corresponds to a metabolic rewiring characterized by a diminished capacity for metabolic plasticity (Fig. 3F). This is notably associated with changes in the levels of key metabolic enzymes like CPT1. We have delved into the discussion of this reduced plasticity in the revised manuscript.

4.- R1 stated: *“The link to lysosomal function and antigen processing is interesting but, in some parts, lacks mechanistic insight. Especially antigen degradation in lysosomes seems to be perturbed upon *Tfam* KO. Additional information about lysosomal acidification, protease content/activity would give more detailed information”.*

In response to R1's valuable feedback, we conducted a comprehensive analysis of lysosome function in activated B cells, aiming to shed light on the functional implications of *Tfam* deletion in this context. Our new data encompasses:

- Lysosomal Acidification and Protease Activity: Building upon R1's suggestion, we now present information about lysosomal acidification alongside protease activity (Cathepsin activity detected using Magic Red) throughout germinal centre (GC) maturation (Fig. 4K, L). This newly acquired data elucidates a distinct defect in lysosomal function remodeling at the pre-GC stage, providing a plausible explanation for the identified GC maturation deficiency in *Tfam* KO mice.

- Transcriptomic Analysis: Leveraging the recently generated RNAseq data, we conducted a targeted analysis to assess the impact of *Tfam* deletion on lysosome function at the transcriptomic level. Our results suggest the presence of retrograde signaling, emanating from mitochondria to the nucleus, shaping lysosome function (Fig. 4M, Supp Fig. 14E).

- In Vitro Experiments: To further substantiate the connection between mitochondrial defects and inhibited lysosomal function, we introduced new in vitro experiments. These experiments involve a comparative analysis of the ability of WT and KO B cells to present OVA protein and OVA peptide. The former necessitates functional lysosomes for presentation, while the latter does not. Our additional data convincingly demonstrates that *Tfam* KO B cells exhibit no impediments in presenting OVAp, reinforcing the correlation between mitochondrial function and lysosomal activity (Fig. 4C, D, Supp. Fig. 14D).

In summary, the incorporation of R1's insightful suggestions has significantly enhanced our understanding of the intricate mechanism through which mitochondria exert control over antigen presentation in B cells. We have incorporated this new data in Fig 4 and discussed these findings throughout the manuscript.

5.- R1 stated: *"Additionally, I don't really understand the point of the Fold change representation and even find it partially misleading. It seems that in vitro B-cells already exhibit the same lysosomal content as stimulated WT B cells. Similarly in the in vivo subsets it seems as if there is already very high lysosomal contents throughout the different stages. Therefore, in my opinion, it is not really correct to say that KO B cells fail to remodel lysosomes upon antigen encounter. Maybe lysosomes are dysfunctional and therefore the increases in lysosomal content is a compensatory mechanism"*

We fully comprehend R1's comment, and upon completing the lysosome characterization (Fig. 4), we are inclined to believe that R1's hypothesis accurately reflects the observed scenario. Our findings suggest that lysosomes exhibit dysfunction, as evidenced by cathepsin activity, and it appears that KO B cells might be augmenting their lysosomal content as a compensatory mechanism. To prevent potential misinterpretations, we have decided to eliminate the Fold Change representation from Figure 4.

6.- R1 stated: *"The finding of the age-associated B cell population in conditional *Tfam* KO animals upon antigen challenge is very interesting. It would be very informative to also look at bioenergetic characteristics, mitochondrial and lysosomal parameters of these ageing associated population"*

We extend our gratitude to R1 for their positive feedback on this discovery. Furthermore, we value R1's suggestion and have conducted the recommended experiments. The additional data now presented demonstrates that this population associated with aging exhibits a metabolic profile and lysosomal content similar to that of GC cells (Fig. 5C, Supp. Fig. 16B, C, D). Moreover, these findings offer insights into how these B cells, lacking *Tfam* and the ability to dynamically adjust their metabolic profile, adapt to compensate for their deficiencies. Please refer to Figure 5.

Minor points:

1) *Does the spleen architecture in *Tfam* B cell -KOs change?*

We have addressed this point and results are included in Supp Fig. 8.

2) *Can the authors at least speculate why KO B cells show an increase in mitochondrial membrane potential??*

We hypothesize that the rise in mitochondrial potential highlighted by R1 in *Tfam* KO B cells may be attributed to impaired ATP synthase activity, particularly under B cell stimulation. Upon

stimulation, *Tfam* KO B cells exhibit a failure to reconfigure their electron transport chain (ETC), as illustrated in Fig. 1E. The oxygen consumption rates (OCR) persist at levels comparable to naïve B cells, even though there is an augmentation in metabolic influx. This disparity could potentially result in an elevation of the mitochondrial membrane potential. It's important to note that this interpretation is speculative at this stage.

3) *What is the Etomoxir concentrations used? 10 μ M?*

We apologize if the initial version of the manuscript was unclear. The concentration of Etomoxir utilized for both SCENITH and Seahorse techniques is 10 μ M, as specified in the Materials and Methods section.

4) *Quinacrine is not a very precise ATP measurement.*

We acknowledge R1's concerns regarding the underutilization of quinacrine, recognizing that it has not been extensively explored thus far. However, the pressing need to assess ATP levels in specific cell populations, especially those with limited cell numbers, and the potential for combining quinacrine with other surface markers underscore its emerging significance in the realm of immunometabolism.

In response to R1's concerns, we conducted a direct comparison between the quinacrine technique (Suppl. Fig. 11B) and the standard luciferase ATP measurement in vitro-stimulated B cells (Figure R1). Our findings reveal a notable similarity between the results obtained using quinacrine and luciferase, with quinacrine demonstrating a capacity to reflect relative changes detected by this more sensitive quantitative technique (Figure R1). While it's acknowledged that the luciferase technique exhibits greater sensitivity, quinacrine proves to be a valid method for assessing the bioenergetic status of cells, albeit with some limitations in sensitivity.

We believe that this validation of quinacrine's reliability may address R1's concerns, and we are open to incorporating this information into the manuscript unless R1 suggests otherwise.

Fig. R1. *Tfam* KO cells demonstrate the ability to enhance ATP production in response to stimulation. The figure displays the relative ATP concentration (nM) in 24-hour cultured wild-type (n = 3) and *Tfam* knockout (n = 2) purified B cells under non-stimulated conditions or stimulated with α -IgM + α -CD40. The values are normalized to non-stimulated wild-type cells. For measurement, 1.5×10^5 B cells were lysed in 100 μ L Tris-EDTA buffer, and 10 μ L of the lysate was utilized for luminescent assay (luciferase reaction).

5) *In the T cell/ B cell coculture experiment -> I don't get the experiment with the soluble stimuli and the conclusion thereafter. As soon as CD3/CD28 is added T cells get activated just by that, no matter if B cells are there and it is completely independent of antigen-presentation by the B cells. Wouldn't a OVA-teramere stain be more relevant for checking antigen-presentation?*

We appreciate R1 for highlighting the need for clarification, indicating that our experimental settings may not have been clearly explained. The stimulation of both B and CD4 T cells with a soluble stimulus was carried out to demonstrate that CD4 T cells lack any intrinsic proliferation defects unless antigen presentation occurs within the context of MHCII by B cells. We have revised this section of the text for clarity. Additionally, in response to R1's suggestion, we have conducted new analyses to assess antigen presentation in B cells. This includes a comparative study examining the impact of presenting OVA protein versus OVA peptide, as mentioned earlier (Supp. Fig. 14). We trust that these adjustments fortify the robustness of our results.

6) *In line 275 authors state that B cells did not show any relevant differences in mTORC1 signaling, However in Fig 3B mature GC show a significant increase in p-S6 levels indicating a perturbed bioenergetic state.*

We extend our appreciation to R1 for their valuable comment, as it has been instrumental in uncovering a correlation between these modifications and shifts in pAKT signaling. This observation suggests a potential impact of mitochondrial function on pAKT signaling, presenting an intriguing avenue for future exploration. We have integrated this novel finding into Supplementary Fig. 11D.

Reviewer #2 (GC responses):

The germinal center (GC) response is key for establishing long-lived humoral immune responses. B cells in GC are highly proliferative and thus metabolically very active. Better insight into the metabolic demand of GC B cells is required, and in particular what the role of mitochondria in this process is. This study reports the generation and analysis of B-cell specific Tfam KO mice. Tfam is a transcription factor for mitochondria biogenesis. Tfam KO mice displayed a blockage of GC reaction, featured by increased differentiation towards memory B cells and aged related B cells, hallmarks of an aged immune response. Unexpectedly, GC blockage in Tfam KO mice did not cause defects in the bioenergetic supply. In contrast, the Tfam KO GC phenotype showed a defect in remodeling the lysosome compartment of B cells.

Overall, I found the paper very interesting, well-written and transparent. The main point is that the conclusion that the GC phenotype in vivo is caused by mitochondrial remodeling is weak/circumstantial. As Tfam plays a crucial role in bone marrow development in the pro-B cell stage, where metabolic checkpoints have been described previously [ref 49], the B cell-specific Tfam KO mice display a severe reduction in the number of splenic B cells. As a consequence, GC formation is expected to be defective and is not necessarily caused by defective mitochondria (explained below).

We thank R2 for her/his encouraging comments. We understand her/his concerns about the impact of low number of B cells and the likely impact in the GC reaction. We have addressed R2's feedback below.

Specific R2 comments:

Major points:

1.- R2 stated: *“Immunization resulted in an impaired GC reaction, which the authors attribute to a failure in mitochondrial remodeling (Fig. 2). Since the KO mice already have very low B cell numbers to start with (Fig. 1A), it is expected that they will have impaired GC formation after immunization. Can the authors exclude that the defective GC response is just caused by the low B cell numbers in the KO mice (due to the block in B cell development in the bone marrow)? In other words, what is the evidence that defective mitochondrial remodeling is responsible for the GC phenotype in vivo?”*

We appreciate R2's input and have given careful consideration to their comments. We have diligently addressed these concerns using different approaches. Notably, we have incorporated the analysis of the *Tfam*^{fl} Cg1^{Cre} model (Supp. Fig. 9) into our study. This particular mouse model facilitates Tfam deletion upon B cell stimulation in the periphery, thereby circumventing the potential influence of a low B cell count. Our findings consistently validate that deficiencies in mitochondrial remodeling during antigen encounters result in impaired germinal center reactions.

2.- R2 stated: *“The same holds true for the data presented in Fig 5. It is clear that Tfam KO mice generate more age-associated B cells upon immunization, but what is the evidence that this is caused by defective mitochondrial remodeling (as the authors conclude in line 421)?”*

We express our gratitude to the reviewer for bringing this to our attention. To establish the intrinsic role of mitochondrial function in aged-associated B cell (ABC) development, we employed two distinct in vitro strategies. Firstly, we compared the differentiation capacity of WT and KO B cells towards ABC upon B cell stimulation, thereby circumventing the potential impact of a low cell count (Fig. 5F). Secondly, we assessed the influence of chemically induced mitochondrial or lysosomal dysfunction on the in vitro induction of ABC (Supp. Fig. 18). We

believe that the inclusion of this new data in the current manuscript provides additional evidence supporting the notion that defective mitochondria intrinsically contribute to aged-associated B cells.

3.- R2 stated: *“What are effects of Tfam KO on circulating antibody levels (naïve vs immunization)?”*

In response to the recommendation from R2, we conducted measurements of circulating antibody levels. To address the issue of low cell numbers in the *Tfam*^{f/f} Mb1Cre system, we utilized the *Tfam*^{f/f} Cg1^{Cre} system for this analysis. The data, presented in Supp. Fig. 15B, underscores the critical importance of mitochondria remodeling in class switch recombination and affinity maturation. We express our gratitude to the reviewer for their valuable suggestion, as we firmly believe that this data is pivotal in substantiating the role of mitochondria remodeling in the maturation of germinal centers, consequently leading to an extrafollicular and aged immune response.

4.- R2 stated: *“The paper would be strengthened if some of the (in vitro) findings were confirmed in human B cells (or human B cell lines). Did the authors attempt to rescue the Tfam KO phenotype in vitro by reintroducing Tfam expression in the cells? (to exclude off-target effects of the Cre-induced deletion)”*

In response to R2's suggestion, we tried to replicate the discovery of mitochondria remodeling influencing lysosomal function in human B cell lines (RL, RAMOS, U2932, SU-DHL4). Initial attempts involved employing CRISPR-Cas9 technology to delete *Tfam* in these cell lines; however, we encountered difficulties in infecting/transfecting these cells, resulting in a complete failure to establish a human cell line that could fully replicate the findings observed in mouse cells. Furthermore, this obstacle prevented us from demonstrating that the reintroduction of *Tfam* expression could rescue lysosomal function.

To address this challenge and gain insights into human cell lines, we resorted to generating p0 Human cells, which lack mitochondrial DNA¹. While we acknowledge that this may not precisely mirror *Tfam* KO cell lines, a comparison of lysosome levels between Control cells and p0 cells revealed alterations in lysosome content. This suggests the presence of a mitochondria-lysosome axis in human B cells (Fig. R2A). We have incorporated these results solely in the point-by-point response document, unless R2 deems it necessary to include this data in the main manuscript.

5.- R2 stated: *“The results shown in Fig 4 are striking and support impaired antigen presentation by Tfam KO B cells. The ‘remodeling’ of lysosomes is presented as the most likely mechanism responsible for the defects in antigen degradation (line 370). However, there are other possibilities to explain the defect in antigen degradation. Is the pH (drop) and expression of cathepsins the same in KO and WT lysosomes? In addition, is the B-T conjugate formation comparable between Tfam KO and WT cells?”*

We appreciate R2's constructive comments on our results. In response, we conducted a more comprehensive analysis to provide deeper insights into the impact of mitochondrial function on antigen degradation:

- Cathepsin Activity Characterization: As detailed in the results section (Fig. 4K, L), the lack of *Tfam* results in defects in cathepsin function, explaining the impaired ability of *Tfam* KO B cells to degrade antigens properly. This observation led us to propose that the elevated levels of the lysosome compartment detected by Lysotracker may serve as a compensatory mechanism for the lysosome function defect. We have incorporated this information into the discussion section of the revised manuscript.

- Coculture Experiments with OVA Protein vs OVA Peptide Coated Beads: To corroborate the finding that *Tfam* KO B cells struggle with antigen degradation, we compared the antigen-presenting ability of WT and KO B cells using OVA protein or OVA peptide-coated beads. The results demonstrated that *Tfam* KO B cells exhibit a disadvantage only when the antigen requires

lysosome function for presentation in the context of MHCII (OVA protein). This data is summarized in Suppl Fig 14D.

- Antigen Presentation Assay using Ea System: To further support the impact of Tfam deletion on lysosomal activity, we implemented another technique involving B cells incubated with beads coated with anti-IgM and Ea peptide, where the antigen does not require lysosome degradation. The presentation of Ea peptide in the context of MHCII was monitored with a specific antibody. Results indicated that WT and KO B cells presented this peptide to the same extent (Fig. R2B). While these results are included in the point-by-point response to manage the document's length, we are open to incorporating this data into the main document if R2 deems it appropriate.

- B-T Conjugate Formation: Despite our confidence in the data obtained with OVAp ruling out the possibility of defects in conjugate formation, we heeded R2's suggestion and evaluated, via FACS, the percentage of conjugates in WT and KO conditions using OVAp and OVA protein (Fig. R2C). Results for this assessment are also included in the supplementary document to prevent further elongation of the already extensive main manuscript.

Minor points:

- *Figure 1: there is a dramatic decrease in total number of B220⁺ cells in Tfam KO spleens, yet the frequencies of FO and MZ cells is comparable to WT spleens. Which subset of B cells is then responsible for the decreased B220⁺ population (transitional B cells, other? Please also include total cell numbers of the B cell subsets in WT and KO spleens (Fig. 1A) for clarity.*

In response to R2's guidance, we conducted a more in-depth phenotyping, incorporating elements such as the characterization of transitional B cells. Additionally, we have provided details about the absolute numbers of cells, and this comprehensive information is presented in Supplementary Figure 1. Our findings indicate that the primary impairment occurs during bone marrow development, as elaborated upon in the manuscript.

- *Fig 1G: OCR is impaired in KO cells after stimulation, what about the glycolysis pathway? This can be done by Seahorse ECAR analysis of WT and KO B cells to support the data in Fig. 3.*

We express our gratitude to R2 for bringing this to our attention. R2 is correct in highlighting the absence of Extracellular Acidification Rate (ECAR) data as a measure of glycolysis in our manuscript. We made this choice considering current knowledge^{2,3}, as ECAR values can encompass non-glycolytic acidification resulting from CO₂ produced by the Tricarboxylic Acid (TCA) cycle, potentially leading to misleading conclusions. Instead, we have incorporated information about ATP production rates by mitochondria and glycolysis (Fig. 1F), what we believe is a valid approach to quantify glycolysis rate. Furthermore, we have now enriched the manuscript with additional details on the glycolysis pathway through Met-flow characterization (Fig. 3F).

- *Please check English grammar throughout the manuscript.*

In response to R2's advice, we have employed an external service to rectify English grammar issues throughout the manuscript.

Fig. R2.- (A) mt-DNA depletion in human B cells leads to changes in lysosome content. Bar charts depict mean fluorescence intensity (MFI) levels of mitochondrial-encoded MT-CO2 (left) and levels of lysotracker green (right) in the human control cell line RL and RL p0 (cells treated with Ethidium bromide for two weeks to deplete mt-DNA). Cells were stimulated with anti-IgM for 24 hours. (B) Deletion of *Tfam* does not result in defects in Ea peptide presentation. On the left, surface expression levels of MHCII:Ea in B cells pre-incubated with Ea peptide and IgM-coated beads for 0 and 5 hours are shown. The middle section presents bar charts illustrating MHCII:Ea mean fluorescence intensity (MFI), while the right section displays total levels of MHCII. (C) *Tfam* deletion has no effect on B-T conjugate formation. The bar chart illustrates the percentage of B-T conjugates (gated in live cells). B cells were incubated for 30 minutes with IgM along with either OVA protein or OVA peptide, followed by a 45-minute co-culture at 37°C to facilitate cell interactions.

Reviewer #3 (Immune ageing, systems immunology):

*Pernichi and colleagues set out to investigate the contribution of the metabolic activity in B cells to the germinal center reaction. The experimental system using mice with *Tfam*-deficient B cells and the experimental design are appropriate and the data are generally convincing. The authors convincingly show that mitochondrial activity in B cells is defunct and germinal center activation is impaired. Surprisingly, the B cells have sufficient metabolic plasticity to compensate for the mitochondrial defects. The authors then convincingly show that the major defect lies in a failure of the *tfam*-deficient to stimulate *Tfh* cells.*

We thank R3 for her/his positive comments regarding our work.

Specific R3 comments:

1. R3- stated: *“Given the huge difference between oxidative phosphorylation and glycolysis in ATP production, Fig. 3a and 3b are surprising. How do the authors explain the greater ATP content in KO mice early on and the increased mTORC1 activity at the mature stage?”*

We consider R3’s question highly pertinent. In this revised version of the manuscript, we have incorporated a more in-depth metabolic profiling of GC maturation using Met-Flow (Fig. 3F). The new data reveal a profound metabolic plasticity in *Tfam* KO B cells. The most significant differences between WT and KO cells are evident in FO and FO^{GL7} cells, indicating a substantial increase in G6PD, CPT1, and IDH1. This suggests an enhanced flux of glucose and fatty acids, compensating for the lack of mitochondrial remodeling. The heightened presence of fatty acids, known to yield more ATP than other substrates, may explain the observed increase in total ATP at these stages (Fig. 3A).

Regarding the noted increase in mTORC1 activation at mat-GC stage, we thank to the R3 for highlighting this aspect. This led us to a more comprehensive characterization of upstream signaling of mTORC1, including the examination of pAKT. Our results indicate that the observed increase in mTORC1 activation during the maturation stage of GC in KO B cells correlates with elevated pAKT levels. This suggests a potentially stronger BCR signaling in these cells, which may contribute to the observed increase in mTORC1. Regardless of the specific mechanism, we find this to be a compelling discovery, opening avenues for future investigations in our laboratory to explore how mitochondrial function directly shapes BCR signaling.

We have tried to discuss about these aspects in the new version of the manuscript.

2. R3 stated: *“The data on failed *Tfh* activation are convincing, however, the evidence for defective antigen presentation is only indirect. Do B cells express MHC class II and costimulatory molecules? Are antigens that do not require processing such as an ova peptide or a superantigen able to stimulate *Tfh* cells?”*

Based on the recommendation from R3, we have examined the expression of specific costimulatory molecules (such as CD40, ICOSL) and MHCII during germinal center maturation. The compiled data is presented in Supplementary Figure 11A. Notably, our analysis revealed a modest yet significant decrease in ICOSL expression at the pre-GC stage in terms of costimulatory molecules. Additionally, we observed a significant increase in MHCII levels in KO B cells, particularly during the early activated stages. These findings have been incorporated into the results and discussion sections.

In response to the recommendation from R3 concerning the suggested experiment involving OVA peptide, we have conducted a comparative analysis as advised. Our experiment focused on evaluating the capacity of *Tfam* KO B cells to present either OVA protein or OVA peptide to CD4. Notably, our findings indicate that *Tfam* KO B cells only exhibit impairment when dealing

with antigens that necessitate lysosomal function, specifically OVA protein. These results have been incorporated into Supplementary Figure 14.

3. R3 stated: *“The observation of the generation of ABC cells is of interest, but it remains unclear how this is related to defective T cell help or mitochondrial dysfunction”.*

We thank this reviewer for the positive comment. Moreover, we align with his assessment. In response, we have introduced fresh in vitro experiments that shed light on the intrinsic repercussions of mitochondria and lysosome dysfunction on ABC differentiation (illustrated in Fig. 5 F and Supp. Fig. 18). These results imply that the observed impact may not be contingent on antigen presentation to Tfh cells. A comprehensive account of these findings has been integrated into the newly added results section of the manuscript.

REFERENCES:

1. Nakahira, K., Haspel, J.A., Rathinam, V.A.K., Lee, S.-J., Dolinay, T., Lam, H.C., Englert, J.A., Rabinovitch, M., Cernadas, M., Kim, H.P., et al. (2011). Autophagy proteins regulate innate immune responses by inhibiting the release of mitochondrial DNA mediated by the NALP3 inflammasome. *Nat. Immunol.* 12, 222–230. 10.1038/ni.1980.
2. Mookerjee, S.A., Gerencser, A.A., Nicholls, D.G., and Brand, M.D. (2017). Quantifying intracellular rates of glycolytic and oxidative ATP production and consumption using extracellular flux measurements. *J. Biol. Chem.* 292, 7189–7207. 10.1074/jbc.m116.774471.
3. Mookerjee, S.A., Goncalves, R.L.S., Gerencser, A.A., Nicholls, D.G., and Brand, M.D. (2015). The contributions of respiration and glycolysis to extracellular acid production. *Biochim. Biophys. Acta (BBA) - Bioenerg.* 1847, 171–181. 10.1016/j.bbabi.2014.10.005.

REVIEWERS' COMMENTS

Reviewer #1 (Remarks to the Author):

The authors have very nicely addressed all of my concerns. I have now further comments.

Reviewer #2 (Remarks to the Author):

The authors have sufficiently revised the manuscript.

Reviewer #3 (Remarks to the Author):

The authors have appropriately addressed the issues I raised. The new data on lysosomal function, antigen presentation and ABC B cell development provide some mechanistic support for the claims, and the overall manuscript is improved. I do not have further comments.